

# Coastal earthquake induced landslide susceptibility during the 2016 Mw 7.8 Kaikōura earthquake, New Zealand

Colin K. Bloom[1], Corinne Singeisen[1], Timothy Stahl[1], Andrew Howell[1,2], Chris Massey[2], Dougal Mason[1,3]

[1]School of Earth and Environment, University of Canterbury, Christchurch, 8014, New Zealand
[2]GNS Science, Lower Hutt, 5011, New Zealand
[3]WSP, Wellington, 6011, New Zealand

*Correspondence to*: Colin K. Bloom (colinkbloom@gmail.com)

**Abstract.** Coastal hillslopes often host higher concentrations of earthquake induced landslides than those further inland, but

few studies have investigated the reasons for this occurrence. As a result, it remains largely unclear if regional earthquake induced landslide susceptibility models trained primarily on inland hillslopes are effective predictors of coastal susceptibility. The 2016 $M_w$ 7.8 Kaikōura earthquake on the northeast South Island of New Zealand resulted in c. 1,600 landslides > 50 m$^2$ on slopes > 15° within 1 km of the coast. This forms an order of magnitude greater landslide source area density than inland hillslopes within 1 to 3 km of the coast. In this study, the distribution of regionally predictive landslide susceptibility variables,

or features, and logistic regression modelling are used to investigate how landslide susceptibility differs between coastal and inland hillslopes and determine the factors that drive the distribution of coastal landslides initiated by the 2016 Kaikōura earthquake. Strong model performance (Area under the Receiver Operator Characteristic Curve or AUC of c. 0.80 to 0.92) was observed across eight models, which adopt four simplified geology types. The same landslide susceptibility factors, primarily geology, steep slopes, and ground motion are strong model predictors for both inland and coastal landslide

susceptibility in the Kaikōura region. In three geology types (which account for more than 90% of landslides source areas) a 0.03 or less drop in model AUC is observed when predicting coastal landslides using inland trained models. This suggests little difference between the features driving inland and coastal landslide susceptibility in the Kaikōura region. Geology is similarly distributed between inland and coastal hillslopes and PGA is generally lower in coastal hillslopes. Slope angle, however, is significantly higher in coastal hillslopes and provides the best explanation for the high density of coastal landslides

during the 2016 Kaikōura earthquake. Existing regional earthquake induced landslide susceptibility models trained on inland hillslopes using common predictive features are likely to capture this signal. Interestingly, in the Kaikōura region, most coastal hillslopes are isolated from the ocean by uplifted shore platforms. Enhanced coastal landslide susceptibility from this event appears to be a legacy effect of past active erosion, which preferentially steepened these coastal hillslopes.





## 1 Introduction

Steep rocky coastlines, which account for c. 80% of waterfront around the globe (Emery and Kuhn, 1982), are a naturally desirable location to live and recreate. A growing global population has resulted in human encroachment on coastlines with nearly one quarter of the human population now living in close proximity to the coast (Small and Nichols 2003). At the same time, global climate change and a rising sea level threaten to increase the rate of coastal landsliding and cliff retreat, in part, due to wave action overtopping protective beaches and impacting coastal cliff faces more frequently (e.g., Young et al., 2014;

Limber et al., 2018). This may have devastating consequences for people and infrastructure near coastal hillslopes (e.g., Jibson, 2006; Dellow et al., 2017; Handwerger et al., 2019).

To help mitigate and manage these hazards, previous studies have attempted to define landslide susceptibility models for steep coastal regions using a physical understanding of the forcings that contribute to coastal mass wasting and the susceptibility

factors that make failure more likely (e.g., Keefer, 2000; He and Beighley, 2008; Budetta et al., 2008; Dickson and Perry, 2016; Francioni et al., 2018; Young, 2018; Limber et al., 2018). Forcings include rainfall, wave and tidal action, and storm surge whereas susceptibility factors include steep topography, geology, rock structure, hydrology, urbanization, and soil moisture among other factors (Van Jones et al., 2015; Dickson and Perry, 2016; He and Beighley, 2008). In tectonically active regions, earthquakes also act as a forcing contributing to the distribution of coastal landslides (Griggs and Plant, 1998; Hancox

et al., 2002) but few coastal models have considered the influence of strong ground motion.

Similarly, while a number of studies (e.g., Budimir et al., 2015; Parker et al., 2015; Massey et al., 2018) have attempted to define factors contributing to regional earthquake induced landslide susceptibility, few have focused specifically on coastlines. In several cases (e.g., Griggs and Plant, 1998; Collins et al., 2012; Massey et al., 2018) a significantly higher landslide density

was observed on coastal hillslopes as compared to inland hillslopes. Given the influence of increased precipitation, weathering, and soil moisture along coastlines, it is possible that regional earthquake induced landslide susceptibility models, which are trained primarily on inland hillslopes, may not effectively predict coastal landslide distributions.

Following the 2016 $M_w$ 7.8 Kaikōura earthquake along the northeast coast of the South Island of New Zealand (Hamling et

al., 2017), Massey et al. (2018) observed an order of magnitude greater number of landslides along coastal hillslopes as compared to inland hillslopes. No clear physical control on landslide density was identified although several hypotheses were explored. Here the distribution of coastal landslides from the 2016 Kaikōura earthquake is used to: 1) compare and contrast the results from earthquake induced landslide susceptibility models developed for coastal and inland hillslopes in the Kaikōura region; 2) evaluate the factors that might contribute to an increased coastal coseismic landslide density during the earthquake;

and 3) explore some of the mechanisms that result in increased coseismic landslide susceptibility along the Kaikōura coast.



## 2 Background

### 2.1 2016 Mw 7.8 Kaikōura Earthquake

The 2016 $M_w$ 7.8 Kaikōura earthquake initiated on the Humps fault near the township of Waiau c. 40 km inland from the coast in the northeastern South Island of New Zealand (Hamling et al., 2017). The earthquake triggered a cascade of fault ruptures

on more than 20 on- and off-shore faults primarily to the northeast of the epicentre (Figure 1, Litchfield et al., 2018). The earthquake ruptured faults of both the North Canterbury and the Marlborough Fault System tectonic domains (Figure 1, Litchfield et al., 2018) and caused complex surface deformation along c. 110 km of coastline (Clark et al., 2017).

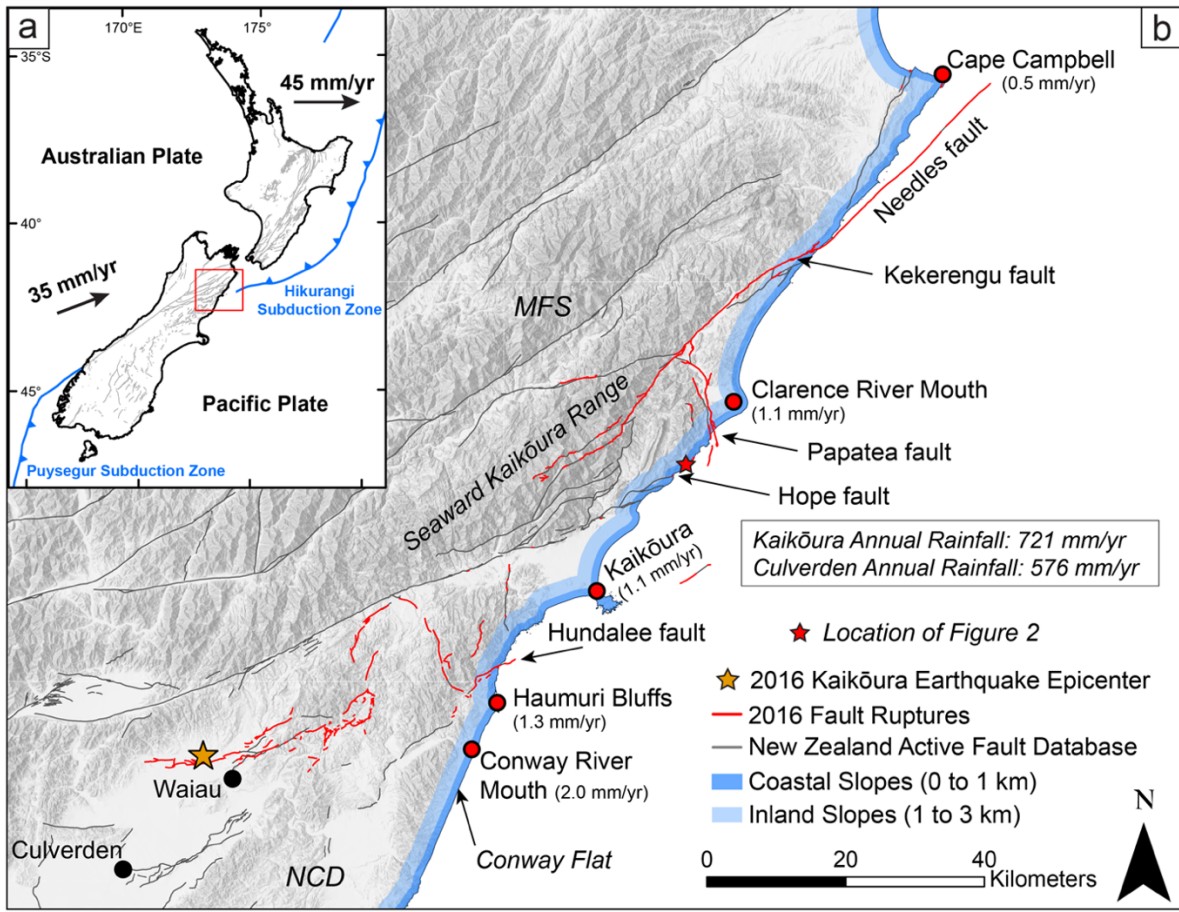

**Figure 1: (a) Location of the Kaikōura earthquake within the tectonic setting of New Zealand. Active faults from the 1:250,000 scale**
**New Zealand Active Fault Database (Langridge et al., 2016) are in grey and simplified major offshore structures are in blue. Black arrows show the relative motion of the Pacific and Australian Plates (Beavan et al., 2002). (b) Area of the Kaikōura earthquake with active faults of the New Zealand Active Fault Database in grey and fault ruptures from the 2016 Kaikōura earthquake in red (Litchfield et al., 2018; Zinke et al., 2019). The shaded blue area is the focus of this study, a 3 km buffer of the coast where modelled PGA from the Kaikōura earthquake was greater than 0.2 g. Red points identify locations where Ota et al. (1996) estimated coastal**
**uplift rates (noted next to labels). Major faults that cross the coastline in the Kaikōura region are labelled. The labelled MFS or Marlborough Fault System is north of the Hope fault and the NCD or North Canterbury Domain is south of the Hope fault. The base image is a multidirectional hillshade of the Land Information New Zealand 8 m DEM (LINZ, 2022).**



The earthquake generated more than 30,000 landslides which were primarily concentrated within the steep slopes of the

Seaward Kaikōura range, around surface fault ruptures, and in steep sections of coastline (Figure 1 and 2; Massey et al., 2018, 2020a; Bloom et al., 2022a). Statistical modelling by Massey et al. (2018, 2020a) found that the regional distribution of landslides from the Kaikōura earthquake was well explained by geology, slope, distance to surface fault traces, peak ground velocity (PGV), local slope relief, and elevation. While Massey et al. (2018) acknowledged a higher density of landslides along the Kaikōura coast, they did not investigate coastal landslide susceptibility, nor the underlying mechanisms involved in the

distribution of coastal earthquake induced landslides. Here, we separate out coastal versus non-coastal hillslopes from this regional analysis and independently investigate the factors that might have contributed towards increased landslide susceptibility of coastal hillslopes during the 2016 Kaikōura earthquake.

**Figure 2: A multidirectional hillshade of post-earthquake lidar (Massey et al., 2020b) with coastal earthquake induced landslides**
**mapped by Massey et al. (2020a). A high density of coastal earthquake induced landslides were observed within the scars of relict landslides that are common along the coastline. Shore platforms modified by road and rail corridors buffer the base of the coastal hillslopes.**



## 2.2 Coastal and Geologic Setting

Much of the Northeast coast of New Zealand's South Island is steep and rocky (Figure 1 and 2). Coastal hillslopes are primarily
composed of intensely jointed Lower Cretaceous greywacke of the Torlesse Supergroup along with younger Upper Cretaceous
to Neogene sedimentary units (Rattenbury et al., 2006). These units are, in places, overlain by less consolidated Pleistocene
alluvial, fluvial, and beach deposits. Portions of the region's steep coastline, primarily south of the Haumuri Bluffs at Conway
Flat (Figure 1), form steep coastal cliffs c. 50 to 150 m in height, many of which are subject to wave action at high tide (Bloom
et al., 2022b). North of the Haumuri Bluffs, however, most coastal hillslopes are uplifted and buffered from direct wave action
by shore platforms that have, in places, been anthropogenically modified to facilitate road and rail corridors (Figure 2; Mason
et al., 2017; Stringer et al., 2021).

Long-term coastal uplift in the Kaikōura region is locally variable as a result of major faults, including the Hope, Kekerengu-
Needles, and Hundalee, which cross-cut the coastline (Figure 1, Litchfield et al., 2018; Howell and Clark, 2022). Approximate
regional estimates of uplift based on Pleistocene marine terraces suggest c. 2.0 mm y$^{-1}$ of uplift at the Conway River mouth,
1.3 mm y$^{-1}$ at the Haumuri Bluffs, 1.1 mm y$^{-1}$ at Kaikōura township, 1.1 mm y$^{-1}$ at the Clarence River mouth, and c. 0.5 mm
y$^{-1}$ c. 10 km south of Cape Campbell (Figure 1; Ota et al., 1996). These measurements generally align well with more recent
measurements of c. 0.9 to 1.3 mm y$^{-1}$ at Kaikōura (Nicol et al., 2022). Single event vertical displacement from the 2016
Kaikōura earthquake ranged from c. −2.5 to 6.5 m along the Kaikōura coast (Clark et al., 2017; Howell and Clark, 2022), but
the areas that subsided in 2016 have undergone net uplift over the Holocene and Pleistocene (Ota et al., 1996; Howell and
Clark, 2022).

As a result of low coastal population density, little work has been done to estimate long term coastal retreat rates for the South
Island of New Zealand. The few studies that have estimated retreat (Bloom et al., 2022b; Kirk 1975, 1977) suggest highly
variable rates modulated by lithology and topography. Average rainfall measured in the township of Kaikōura is c. 721 mm
yr$^{-1}$ (Macara, 2014) but is highly spatially variable along the Kaikōura coast ranging from c. 675 to c. 1500 mm yr$^{-1}$ (NIWA,
2022). Over the historical record, significant landsliding along the Kaikōura coast has been observed following large storms,
for example Cyclone Alison (March 1975) and ex-Tropical Cyclone Ita (April 2014), which brought high rainfall to the region
over a short period of time (Massey et al., 2021a).

## 3 Data and Methods

To explore coastal earthquake induced landslide susceptibility in the Kaikōura region, we rely on a combination of predictive
landslide susceptibility features and an earthquake induced landslide inventory produced by Massey et al. (2020a) following
the 2016 Kaikōura earthquake (Figure 2). These datasets are used to examine the distribution of landslides with distance from
the Kaikōura coast and provide training data for comparative inland and coastal landslide susceptibility models (Figure 3). For



the purposes herein, inland hillslopes are defined as hillslopes within >1 to 3 km of the Kaikōura coast and coastal hillslopes
as hillslopes within 1 km of the coastline (Massey et al., 2018). This distinction was based on three main factors. First, the
observed landslide density is much higher within 1 km of the coast than within 1 to 3 km despite generally similar distributions
of lithology, elevation, and vegetation. Second, hillslopes greater than 3 km from the coast capture a different proportion of
lithology and alpine terrain (Figure 1) with a higher landslide density. These hillslopes are more difficult to compare with the

coastal setting and may be affected by different processes. Third, the area within 1 km of the coast primarily encompass terrain
up to the first topographic ridgeline (Massey et al., 2018), and is therefore more representative of 'coastal-facing' hillslopes
than those at greater distances. The differences between the coastal and inland susceptibility models are investigated to examine
the factors contributing to an increased density of landslides on the Kaikōura coast. The following sections provide an overview
of these datasets and analyses.

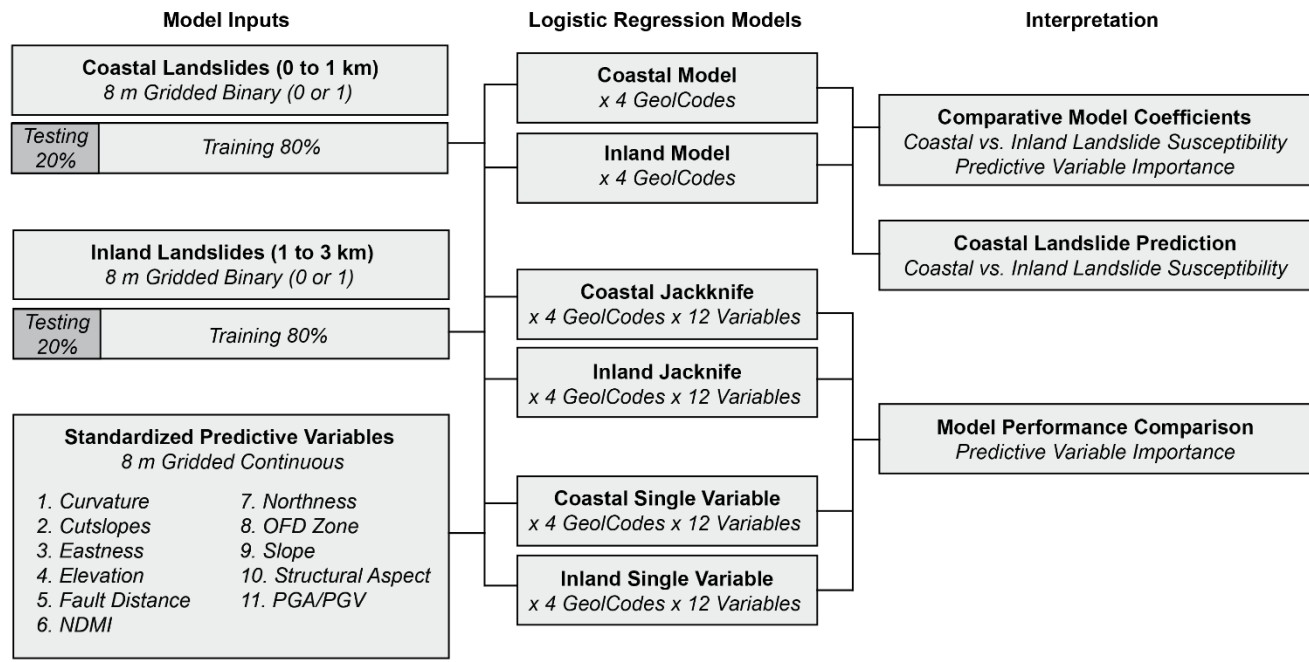


**Figure 3: Logistic regression model workflow. Gridded landslide data and predictive features are used to train and test binary logistic regression models. Model results are used to compare inland and coastal landslide susceptibility and assess the importance of predictive features. NDMI stands for Normalized Difference Moisture Index while PGA/PGV stands for Peak Ground Acceleration/Velocity.**

**3.1 2016 Kaikōura Landslide Inventory**

For this analysis, mapped source area polygons from version 2.0 of the 2016 Kaikōura earthquake induced landslide inventory
(Figure 2 and 3; Massey et al., 2020a) were converted into a binary 8 m grid of landslide and non-landslide grid cells. Grid
cells were assigned a value of 1 if the centre point of the grid cell fell within a landslide source area polygon and a value of 0
if it did not. Based on the size area distribution of landslides within 3 km of the Kaikōura coastline, landslides smaller than 50




$m^2$ were excluded from the analysis. This was an effort to eliminate potential bias resulting from preferential mapping of smaller failures along the Kaikōura coastal road corridor (Figure A1). Here, distance to the Kaikōura coastline was defined using the nearest Euclidean distance from the centre point of each 8 m grid cell to the coast as defined by the Land Information New Zealand Topo50 coastline (LINZ, 2022). Investigations were limited to slopes greater than 15° which captured c. 91% of cumulative landslide source area while excluding most hillslopes unlikely to produce significant landsliding in this region.

This threshold is commonly applied (e.g., Meunier et al., 2007; Kritikos et al., 2015) to confine investigations to hillslopes capable of producing landslides. Furthermore, analyses were limited to those areas where modelled PGA (Worden et al., 2020) was greater than 0.2 g (the triggering threshold for New Zealand's earthquake induced landslide response tools; Massey et al., 2021b). Areas with PGA greater than 0.2 g include c. 99% of mapped coastal landslide source areas from the 2016 Kaikōura earthquake. Finally, the c. 452,000 $m^2$ Seafront landslide was excluded from the analysis to avoid skewing descriptive statistics

and modelling. The source area of the Seafront landslide occurs approximately 1 to 3 km from the coast but is located directly along a surface-rupturing fault (>9 m vertical displacement; Bloom et al., 2021). The failure is almost an order of magnitude larger than the next largest 2016 failure within 3 km of the coastline and is not representative of failures within the wider coastal region.

To supplement the landslide inventory, we reviewed the high resolution pre- and post-event digital elevation models and orthophotographs used to create the Kaikōura earthquake induced landslide inventory (Massey et al., 2020a) and manually assigned each landslide source area polygon either a 'First Movement' designation or one of three landslide activity designations derived from Cruden and Varnes (1996): 'Reactivated Retrogressive Rock or Debris', 'Reactivated Moving Debris', or 'Reactivated Moving Rock'. These designations represent the landslide activity in relation to past failures. For the

purposes herein, a past failure was defined as any landslide, landslide debris, or landslide scar that was apparent on the hillslope before the 2016 earthquake. First movements have no obvious link with a pre-2016 failure. Reactivated Retrogressive Rock or Debris exhibited extension of a pre-2016 landslide head scarp opposite to the failure direction. In this case we also include coastal cliff-top failures where there was evidence of past landslides. Reactivated Moving Rock exhibited remobilisation of the majority of material within an observed pre-2016 landslide. Reactivated Moving Debris exhibited minor deformation

within the body of a past landslide including partial reactivation of landslide debris.

### 3.2 Landslide Distribution

To examine the distribution of landslides in relation to the coast, the landslide source area density, referred to herein as landslide density, was calculated at increasing distance from the coast. Landslide density is the sum of gridded landslide source area within 24 m bins at distance from the 1:50k Topo50 New Zealand Coastline (LINZ, 2022) divided by the total area within

the bin. Landslide density was further broken down within five geology types (GeolCodes) simplified from the New Zealand QMAP (Rattenbury et al., 2006). GeolCode 1 represents Quaternary sands, silts, and gravels that are primarily fluvial deposits but also include alluvial fan, marine, and recent beach deposits; GeolCode 2 represents Neogene limestones, sandstones, and



siltstones; GeolCode 3 represents Upper Cretaceous to Paleogene rocks including limestones, sandstones, and siltstones, GeolCode 4 represents minor undifferentiated volcanic rocks; GeolCode 5 represents Lower Cretaceous Torlesse (Pahau

terrane) basement rocks that are predominantly heavily deformed sandstones and argillite, commonly referred to as greywacke in New Zealand; and finally GeolCode 6 represents undifferentiated relict landslides and hillslope deposits as defined by QMAP. It is important to note that relict landslides and hillslope deposits are not systematically mapped within QMAP (Rattenbury et al., 2006).

### 3.3 Landslide Susceptibility Features

A number of machine learning and statistical modelling techniques, for example logistic regression, random forest, or deep neural networks, have been successfully applied to regionally estimate landslide susceptibility (Reichenbach et al., 2018). The choice of modelling technique is largely governed by the scale and requirements of the analysis. In this study the binary logistic regression technique is used because it 1) balances relatively high model performance with low model training times and 2) has high 'explainability' allowing us to easily identify the importance of individual predictive features in trained models

(Figure 3; Budimir et al., 2015). The purpose of using machine learning in this study is to better understand predictive features not to create a forecast model or susceptibility maps. Other types of models, for example deep neural networks, may result in higher model performance more suited to forecasting but sacrifice explainability and require longer training times (Reichenbach et al., 2018) limiting their application in this case.

The basic requirements for all empirical landslide susceptibility analyses are, typically, a categorical landslide dataset (defining the presence or absence of a landslide at any given location) and one or more predictive features that can be used to train the model (Figure 3; Budimir et al., 2015). A separate categorical landslide dataset (not used in model training) is used to test the efficacy of the trained model (Figure 3). Model performance is optimised differently based on the modelling technique but usually involves varying model 'hyperparameters' or values used to control model process, and refining the predictive features

used to train the model (Lombardo and Mai, 2018; Reichenbach et al., 2018). Here, we discuss the choice of predictive features and further describe the application of the logistic regression modelling technique.

For this analysis, 25 common predictive features used in other landslide susceptibility studies (e.g., Budimir et al., 2015) were developed. These features included a range of lithologic, topographic, and surface conditions, for example, slope angle,

roughness, and vegetation greenness (NDVI) (Figure A2). Of these features, 13 that produced variable inflation factor (VIF) scores greater than 10 were excluded. Removal limits the influence of collinear features, generally improving model performance, and maintaining model explainability (Lombardo and Mai, 2018).

The various features used in this analysis (Table 1) were converted to raster format and/or aligned to the spatial resolution of

the LINZ 8 m DEM (LINZ, 2022) using the GDAL Rasterize and Warp functions with nearest neighbour resampling. The



LINZ 8m DEM, primarily derived from the January 2012 LINZ Topo50 20m contours (LINZ, 2022), was used to derive the curvature, aspect, elevation, and slope features used in the analysis (Table 1).

**Table 1: Landslide susceptibility features.**

| Number | Features | Type | Description |
|--------|----------|------|-------------|
| 1 | Coast Distance | Continuous | Euclidean distance (GDAL) from the 1:50k Topo50 New Zealand Coastline (LINZ, 2022) |
| 2 | Curvature | Continuous | Curvature (RichDEM) derived from Land Information New Zealand (LINZ) 8 m DEM (LINZ, 2022) |
| 3 | Cutslopes | Categorical | Modified cutslopes mapped as polygons and gridded to 8 m. Values of 1 indicate the presence of a cutslope at the grid cell centre point. |
| 4 & 9 | Aspect (Eastness and Northness) | Continuous | Eastness (Sin) and Northness (Cos) of Aspect (GDAL, Converted to Radians) produced using LINZ 8 m DEM (LINZ, 2022) |
| 5 | Elevation | Continuous | LINZ 8m DEM (LINZ, 2022) primarily derived from January 2012 LINZ Topo50 20m contours (LINZ, 2022); Native Resolution: 8 m/pixel |
| 6 | Fault Distance | Continuous | Euclidean distance (GDAL) from 14 surface ruptured faults from 2016 by Bloom et al. (2022a) |
| 7 | Ground Motion | Continuous | PGA and PGV from USGS ShakeMap v4 (Worden et al., 2020); Native Resolution: 336 m/pixel |
| 8 | NDMI (Normalized Difference Moisture Index) | Continuous | Derived from October 2016 Landsat 8 Imagery (U.S. Geological Survey 2022): NDMI = (Band 5 – Band 6) / (Band 5 + Band 6); Native Resolution: 30 m/pixel |
| 10 | OFD (Off Fault Deformation) | Categorical | OFD zone as defined for 14 faults by Bloom et al. (2022a) as polygons gridded to 8 m. Values of 1 indicate the presence of an OFD zone at the grid cell centre point. |
| 11 | Slope | Continuous | Slope (GDAL) derived from LINZ 8 m DEM (LINZ, 2022) |
| 12 | Structural Aspect | Continuous | Difference between aspect and dip direction of QMAP bedding measurements (Rattenbury et al., 2006; see Appendix A for additional detail) |
| 13 | Geology (GeolCode) | Categorical | Simplified from 1:250k scale New Zealand QMAP (Rattenbury et al., 2006) Classes: 1. Quaternary Sands and Gravels, 2. Neogene Sediments, 3. L. Cretaceous–- Paleogene Sediments, 4. Volcanics, 5. Torlesse Greywacke (Pahau), 6. Landslide and Hillslope Deposits |


Similar to the landslide density analysis, the extent of features (Table 1) was limited to slopes greater than 15° and areas with PGA (defined by the USGS ShakeMap; Worden et al., 2020) greater than 0.2 g. Continuous landslide susceptibility features (Figure 3, Table 1) were scaled using the standard scalar method:

$$z = \frac{x-\mu}{\sigma},$$ (1)

where the standardised value ($z$) is the original value ($x$) minus the mean ($\mu$) of all values divided by the standard deviation ($\sigma$) of all values. Using the standard scalar allows us to compare model coefficients, or the weights assigned to each feature during model training, side-by-side.





### 3.4 Logistic Regression Modelling

The predictive power of individual landslide susceptibility features during the Kaikōura earthquake was strongly modulated

by geology type (Massey et al., 2018, 2020a; Singeisen et al., 2022). Separate coastal (0 to 1 km from the coast) and inland (1 to 3 km from the coast) models were, therefore, trained in four simplified geology types (GeolCodes 1, 2, 3, and 5; Figure 3) using predictive features and the sci-kit learn python library (Table 1). GeolCode 4 (Volcanics) and GeolCode 6 (Mapped Hillslope Deposits) lacked sufficient data to support a robust model and were excluded from the analysis. Additional models were trained using the combined data from inland and coastal hillslopes and these results are included in Appendix B (Figures

B1 and B2).

Models were trained on 80% of gridded data leaving 20% of data for independent verification of the model performance (Figure 3). Across model training random 10-fold cross validation was used to evaluate model uncertainty. In K-fold cross validation, the training dataset is partitioned into K (in this case 10) parts (Hastie et al., 2017) and models are iteratively trained

using all parts minus one. The remaining portion of data excluded from training in each iteration is used to validate model performance. An L1 regularization was used to penalize poor features and improve model prediction by simplifying the model (Lombardo and Mai, 2018). Using the L1 (also known as the Least Absolute Shrinkage Selection Operator or LASO; Tibshirani, 1996) allows the model to assign overly collinear or unsupportive features a coefficient of zero. The SAGA solver (Defazio et al., 2014), which supports L1 regularization, was used to weight coefficients. In all cases, models converged prior

to a maximum 100 iterations. Based on hyperparameter tuning, a C (inverse of regularization strength) of 1 was applied to the models. The target datasets have a greater number of non-landslide source area (value of 0) grid cells than landslide (value of 1) grid cells (Table A1). To limit overprediction, no attempt was made to balance or otherwise weight the datasets during model training.

The intention of this work was not to systematically evaluate or compare model prediction. However, estimates of area under the receiver-operator characteristic curve were used to demonstrate the relatively high performance of all trained models. The receiver operator characteristic (ROC) curve (e.g., Fawcett 2006; Lombardo and Mai 2018) plots the true positive rate (TPR) against the false positive rate (FPR) at different probability thresholds. TPR, also known as 'sensitivity,' represents the ratio of positive predictions that were correctly classified as positive by the model, i.e. pixels modelled as failures that actually failed

in 2016. FPR is calculated as (1 – specificity), where specificity is the true negative rate (TNR) or the ratio of negative model predictions that were correctly classified as negative. The shape of the ROC curve is used to evaluate the goodness of fit for a binary classifier – in this case, whether a grid cell represents a landslide source area or not (Y = 1 or Y = 0). The class prediction for each instance is determined based on the probability threshold. Area under the ROC curve (AUC) is calculated to quantify the shape of the curve in a single reportable value. Values of AUC close to 1 represent better model performance while values

close to 0.5 represent near random results (Hosmer et al., 2013). As a final test to demonstrate the efficacy of the models, we





used the results of models trained on inland data to predict the coastal landslide distribution (Figure 3) and this is also reported based on AUC.

Because model features were standardised using the standard scalar method, model coefficients can be directly compared for each predictive feature to estimate feature importance (Figure 3). Following the techniques of Lombardo and Mai (2018) and Williams et al. (2021), jackknife and single variable logistic regression model permutations were also trained for inland and coastal hillslopes in GeolCodes 1, 2, 3, and 5 to further assess feature importance (Figure 3). In the jackknife method, a single landslide susceptibility feature is iteratively removed during model training (Lombardo and Mai, 2018; Williams et al., 2021). Individual model results are then compared to evaluate the influence of removing each feature from the model. A more substantial drop in model AUC suggests higher importance for the removed feature. In the single variable method, models are iteratively trained on each susceptibility feature separately to determine individual feature importance. In these models, a higher model AUC suggests that the feature has a greater independent explanatory value.

## 4 Results

### 4.1 Distribution of Coastal Earthquake Induced Landslides

Similar to the results of Massey et al. (2018), an order of magnitude greater earthquake induced landslide density was observed across coastal hillslopes as a result of the 2016 Kaikōura earthquake (Figure 4). Within 1 km of the coast, 1,621 landslides > 50 m$^2$ were observed on slopes greater than 15° with a mean PGA greater than 0.2 g. Given these filters, on average, coastal landslides were slightly larger than inland landslides (c. 870 m$^2$ for coastal hillslopes and c. 780 m$^2$ for inland hillslopes; Table B1). Removing landslide size and slope filters results in a slightly higher coastal landslide density. Source area density peaks at c. 7% between 0 and 100 m from the coastline and drops to c. 0.5% at 1000 m from the coastline (Figure 4). Between 1000 m from the coastline and 3000 m from the coastline, landslide source area density remains generally consistent with an average density of c. 0.5% (Figure 4).







**Figure 4: Overall landslide source area density (landslide density) within 24 m bins at increasing distance from the Kaikōura coast as defined by the LINZ Topo50 Coastline (LINZ, 2022). Landslide density within GeolCodes 1, 2, 3, and 5 are presented separately in the bottom plots.**





## 4.2 Distribution of Lithology

Most landslides from the 2016 Kaikōura earthquake occur within Torlesse greywacke (GeolCode 5), younger sedimentary units (GeolCode 2 and 3), and unconsolidated Quaternary units (GeolCode 1) (Table 2). Less than 1% of landslides were observed within volcanic rocks (GeolCode 4) or mapped pre-existing failures and hillslope deposits (GeolCode 6, which are not systematically mapped). Regional landslide density does not mirror the distribution of lithology (Table 2) and landslide source areas disproportionately occur within Upper Cretaceous to Paleogene sediments (GeolCode 3) and, along the coast, within Lower Cretaceous Torlesse greywacke (GeolCode 5). The general landslide density trends are primarily driven by these two geology types. Within c. 100 m of the coast, landslide density is as high as c. 6% in both GeolCodes 3 and 5 (Figure 4).

**Table 2: Distribution of lithology and landslides**

| GeolCode | Geology | Percent of Coastal Area (0 to 1 km) | Percent of Coastal Landslides | Coastal Landslide Density | Percent of Inland Area (1 to 3 km) | Percent of Inland Landslides | Inland Landslide Density |
|---|---|---|---|---|---|---|---|
| 1 | Quaternary | 7.3% | 9.0% | 2.5% | 3.8% | 2.1% | 0.3% |
| 2 | Neogene | 32.9% | 20.1% | 1.2% | 21.2% | 20.1% | 0.5% |
| 3 | Paleogene | 16.8% | 25.3% | 3.1% | 23.8% | 39.6% | 0.8% |
| 5 | Torlesse | 42.9% | 45.6% | 2.2% | 51.2% | 38.1% | 0.4% |

## 4.3 Landslide Activity

Of the mapped landslides from the 2016 Kaikōura earthquake, c. 13% within 1 km of the coast and c. 34% between 1 and 3 km of the coast were first movements (Table 3). The remaining failures were a combination of reactivation of relict landslides, including retrogression of pre-existing landslide head scarps and reactivation of landslide debris. Within Torlesse greywacke (GeolCode 5), c. 49% of inland landslides were first movements as compared to c. 17% of coastal failures (Table 3).

**Table 3: Earthquake induced landslide activity**

| Coast (0 to 1 km) | | | | | |
|---|---|---|---|---|---|
| GeolCode | Geology | First Movement | Reactivated Retrogressive Movement | Reactivated Moving Rock | Reactivated Moving Debris |
| 1 | Quaternary | 12% | 43% | 3% | 42% |
| 2 | Neogene | 8% | 54% | 4% | 35% |
| 3 | Paleogene | 13% | 33% | 4% | 50% |
| 4 | Volcanics | 0% | 0% | 0% | 0% |
| 5 | Torlesse | 17% | 28% | 4% | 51% |
| 6 | Relict Landslides (QMAP) | 0% | 39% | 1% | 60% |
| **All** | **All** | **13%** | **38%** | **4%** | **45%** |
| Inland (1 to 3 km) | | | | | |





| 1 | Quaternary | 28% | 39% | 3% | 30% |
|---|---|---|---|---|---|
| 2 | Neogene | 21% | 53% | 4% | 23% |
| 3 | Paleogene | 24% | 40% | 4% | 32% |
| 4 | Volcanics | 31% | 46% | 8% | 15% |
| 5 | Torlesse | 49% | 19% | 2% | 29% |
| 6 | Relict Landslides (QMAP) | 28% | 40% | 4% | 28% |
| **All** | **All** | **34%** | **38%** | **3%** | **25%** |

## 4.4 Coastal vs. Inland Earthquake Induced Landslide Susceptibility Models

AUC of cross-validated coastal models is generally consistent with similar studies (e.g., Reichenbach et al., 2018; Williams et al., 2021) and ranges from c. 0.79 in coastal Neogene sediments (GeolCode 2) to c. 0.92 in coastal Quaternary sediments (GeolCode 1) with generally low variability across 10 cross-validations (Figure 5). Additionally, all model AUCs were within the range of cross-validations when independently testing model performance using 20% of data withheld from model training (Figure 5).

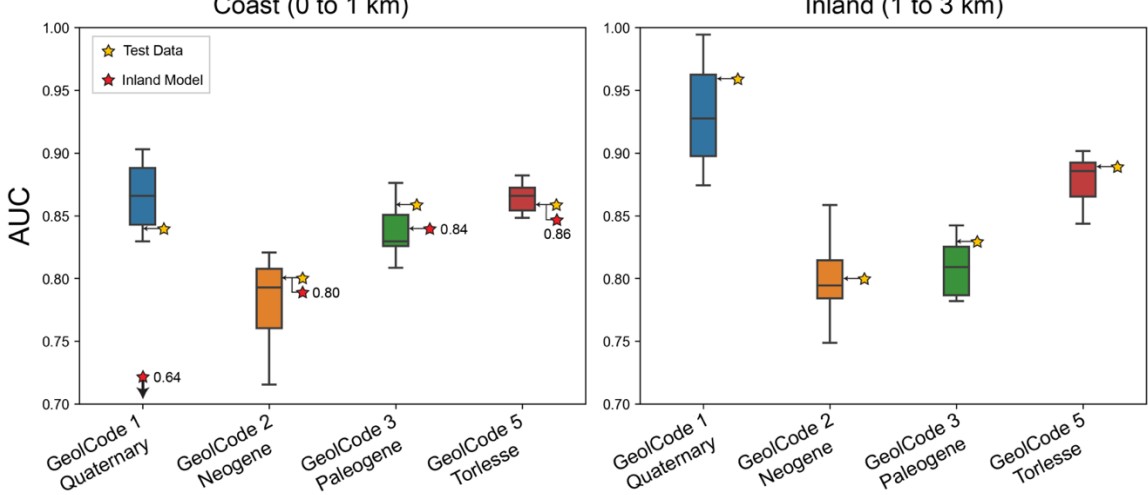

**Figure 5: Logistic regression model performance from models trained on each geology type (GeolCode) in coastal (left) and inland (right) hillslopes. Model performance is measured by area under the receiver operator characteristic curve (AUC). Each boxplot shows the results of 10-fold cross validation using 80% of the available target dataset. Yellow stars represent model performance when applied to the 20% of data withheld from training. Red stars in the Coast results represent the performance of the inland model when applied to the coast dataset. The red star with an arrow pointing down in Coast GeolCode 1 represents an AUC beyond the extent of the plot at 0.64.**

The results of inland model training (Figure 5) were used to predict the coastal landslide distribution. Models trained on inland landslides and applied to coastal hillslopes generally produced the same or lower AUC values than models trained on coastal





hillslopes (Figure 5). There was an c. 0.20 drop in AUC in GeolCode 1, a c. 0.03 drop in GeolCode 3, and almost no drop in GeolCodes 2 and 5 (Figure 5).

Below, we compare and contrast model coefficients alongside the results of jackknife and single variable models for each
320    geology type (Figure 6 and 7) to further examine the relative importance of the predictive features.

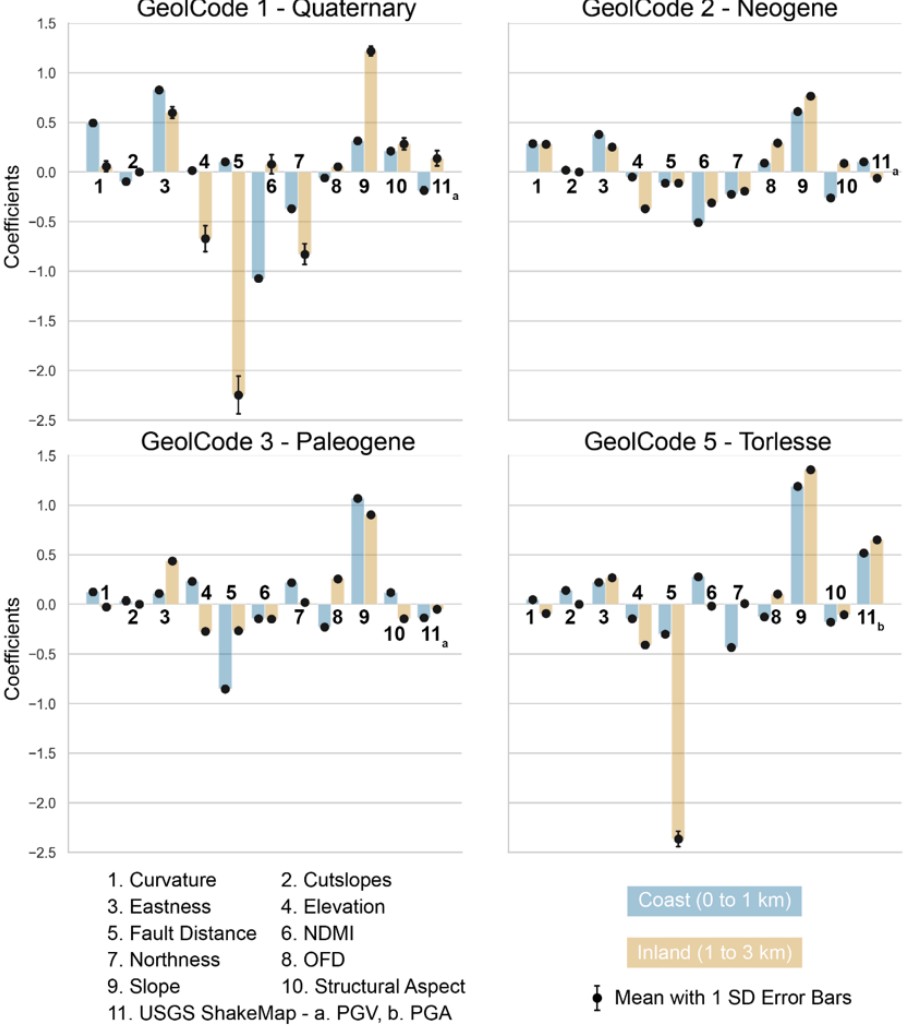

**Figure 6: Model coefficients for models trained on each GeolCode. Points with associated error bars represent the mean and standard deviation (SD) of the coefficient across 10-cross validations. Colour bars indicate which feature the point is associated with and the model data used to train the model (either coast or inland). In cases where no error bar is present, the standard deviation is**
325   **less than 0.1. Negative coefficients result in a higher weight for small values while positive coefficients result in a higher weight for high values. For example, a negative coefficient for fault distance suggests that there is a higher landslide susceptibility closer to faults while a positive coefficient for slope suggests that a greater slope angle has higher landslide susceptibility. All features are standardised prior to model training allowing for the direct comparison of coefficients within the same model.**




**Figure 7: Results of jackknife (left) and single variable (right) logistic regression models. Model performance is measured by area under the receiver operator characteristic curve (AUC). The box plot for each model shows the results of a random 10-fold cross validation. In jackknife models, a single model feature is iteratively removed from model training and a bigger drop in AUC represents higher feature importance. In single variable models, a separate model is trained using each feature and higher AUC represents higher explanatory value. AUC close to 0.5 represents near random results while AUC near 1 represents near perfect results (Hosmer et al., 2013).**

### 4.4.1 GeolCode 1- Quaternary

For inland (1 to 3 km from the coast) unconsolidated Quaternary units (GeolCode 1), distance to fault and slope features had the highest model coefficients (Figure 6). For coastal hillslopes (0 to 1 km from the coast), a low model coefficient was observed for the NDMI (soil moisture) feature suggesting an inverse relationship where lower values of the soil moisture proxy predict higher landslide susceptibility. An c. 0.04 drop in AUC was observed for coastal jackknife models trained without the



NDMI predictor (Figure 7). In single variable models, NDMI alone produced an AUC of c. 0.79 +/- 0.03 (+/- 1 σ) in coastal hillslopes, c. 0.07 higher than the slope-only model which yielded the next highest AUC. In jackknife models of inland hillslopes, there was not a substantial drop in AUC for any model iteration and in single variable models of inland hillslopes, a number of features produced high model performance (Fault distance an AUC of c. 0.88 +/- 0.04, ShakeMap PGV an AUC of c. 0.79 +/- 0.03 and slope an AUC of c. 0.74 +/- 0.11; Figure 7).

### 4.4.2 GeolCode 2 - Neogene

In Neogene sediments (GeolCode 2), a similar distribution of coefficients for inland and coastal hillslopes were observed with the highest model coefficients for the slope feature (Figure 6). Additionally, negative coefficients for NDMI were observed in both inland and coastal hillslopes. While observations of model coefficients were largely supported by jackknife models in both inland and coastal hillslopes, the most substantial drop in AUC (c. 0.05) was seen with the exclusion of the coastal slope feature (Figure 7). Single variable models showed an AUC of c. 0.72 +/- 0.02 for coastal slope and c. 0.75 +/- 0.03 for inland slope features (Figure 7).

### 4.4.3 GeolCode 3 - Paleogene

In Paleogene sediments (GeolCode 3) a similar distribution of coefficients was again observed in inland and coastal hillslopes with the highest model coefficients for the slope and distance to fault features (Figure 6). A strong negative coefficient was also observed for mean PGA in inland hillslopes. In jackknife models there was an c. 0.13 drop in both coastal and inland model AUC with the removal of the slope feature and an c. 0.03 drop in coastal model AUC with the removal of the fault distance feature (Figure 7). In single variable models, slope showed the best model performance with an inland model AUC of c. 0.77 +/- 0.03 and a coastal model AUC of c. 0.81 +/- 0.03 (Figure 7).

### 4.4.4 GeolCode 5 - Lower Cretaceous

In Lower Cretaceous Torlesse greywacke (GeolCode 5) high model coefficients were observed for the slope and mean PGA features though these are strongly outweighed by the fault distance feature in inland hillslopes (Figure 6). An c. 0.09 drop in coastal model AUC and a c. 0.13 drop in inland model AUC was observed with the removal of the slope feature in jackknife models (Figure 7). Interestingly, only an c. 0.01 drop in inland model AUC was observed with the removal of the fault distance feature despite a high model coefficient. As a single feature, slope had the highest AUC in both inland (0.85 +/- 0.2) and coastal (0.82 +/- 0.02) models (Figure 6) while PGA had an AUC of c. 0.72 +/- 0.02.



## 5 Discussion

### 5.1 Logistic Regression Models

Despite an order of magnitude higher landslide density observed within 1 km of the Kaikōura coast (Figure 4), few significant differences were observed between modelled coefficients in inland and coastal landslide susceptibility models. Additional models trained on both coastal and inland data yielded similar model coefficients (Figure B2). Models trained on data from 1 to 3 km inland and applied to coastal hillslopes from 0 to 1 km were still highly predictive and only resulted in a 0.03 or less drop in AUC as compared to models trained and tested on coastal hillslopes in GeolCodes 2, 3, and 5 (Figure 5). These three

geology types account for greater than 90% of coastal landslide density (Table 2).

The larger variation in model performance (c. 0.20) between inland and coastal models of GeolCode 1 could represent a true difference between inland and coastal landslide susceptibility. Inland GeolCode 1, however, accounts for less than 4% of total inland area and c. 2% of inland landslides (Table 2). Given a slightly larger spread in AUC (Figure 5 and 7) and model

coefficients (Figure 6) across 10 cross-validations, it is also possible that there is simply not enough data to train an effective model in inland GeolCode 1.

Some minor differences in model coefficients were observed, in particular the higher importance of fault distance in coastal GeolCode 3 and inland GeolCode 5, but these do little to explain the overall landslide density trend. Despite a high model

coefficient for fault distance in inland GeolCode 5, these was only an c. 0.01 drop in AUC in jackknife models suggesting a potentially high correlation with other predictive features (likely PGA; Figure A2). Across jackknife and single variable models, slope and, in the case of GeolCode 5, PGA appear to be much stronger and more effective features than fault distance for predicting the regional landslide distribution within both inland and coastal hillslopes of the Kaikōura region.

### 5.2 Factors controlling increased coastal landslide density

Modelling of landslide susceptibility successfully captures the coastal distribution of landslides from the Kaikōura earthquake but does not provide a clear explanation for the order of magnitude difference in inland and coastal landslide density. To better explain this occurrence, the distribution of several of the most important landslide susceptibility features from the modelling were further examined (Figure 8; additional features are discussed in Appendix B).





**Figure 8: Landslide density for each GeolCode (black line) within 24 m bins at increasing distance from the Kaikōura coast plotted alongside the distribution of standardised predictive features (orange, green, and red lines) and landslide (LS) susceptibility (blue line) based on a logistic regression model trained on inland data from 1 to 3 km. Standardised features and landslide susceptibility are presented as the mean of values within 24 m bins with distance from the coast.**



*Slope* – Massey et al. (2018) noted a lower overall distribution of slope near the coast in the Kaikōura region. When slopes below 15° are excluded, however, there is a steep rise in slope within c. 500 m of the coastline (Figure 8) that correlates well with the distribution of landslide density across geology types (Figure 4 and 6). Model coefficients and jackknife models (Figure 6 and 7) suggest that slope is one of the most important features determining the distribution of landslides from the Kaikōura earthquake. As such, it is likely that the higher density of landslides observed along the Kaikōura coast is, to a large

extent, a product of this feature.

*Strong Ground Motion (Mean PGA and Distance to Fault)* – Across geology types a decrease in ground motion and increase in fault distance is observed at c. 500 m from the coast. This is particularly evident in GeolCode 5 (Figure 8) and does little to explain the observed landslide density trends. It is important to note, however, that there is a large concentration of landslides

on the northern Kaikōura coast where modelled ground motion is high and model coefficients suggest that, particularly for GeolCode 5, high modelled PGA is a good predictor of coastal landslide density (Figure 6). The steep decrease in modelled PGA/PGV observed near the coast (Figure 8) could be a result of increasing distance from the seismic source south of Kaikōura. Here, coastal landslides concentrate within weaker actively eroded lithologies that fail at lower ground motions (Bloom et al., 2022b).


Topographic and site amplification of seismic waves (e.g., Ashford et al., 1997) likely contributed to local variability in strong ground motion intensity within individual coastal and inland hillslopes during the Kaikōura earthquake. Ground motion variability is known to influence landslide susceptibility (e.g., Massey et al., 2022) but remains challenging to estimate on a regional scale. Outside of applying regional ground motion intensity estimates (PGA/PGV from the USGS ShakeMap), this

analysis does not investigate the role of site-specific ground motion. Given the coarse native resolution of PGA/PGV estimates from the USGS ShakeMap (336 m/pixel; Worden et al., 2020), uncharacterised ground motion variability may have an influence on the distribution of landslides from the 2016 Kaikōura earthquake.

*Lithology and Geologic Structure (Geology and Structural Aspect)* – A similar distribution of lithology was observed in both

inland and coastal hillslopes (Table 2), and it is assumed that, over short distances, geology has a relatively consistent influence on landslide susceptibility. As a result, while geology appears to strongly modulate landslide density, it does not readily explain the increase in coastal landslide density from the 2016 Kaikōura earthquake. Likewise, the correlation between lithologic bedding and topographic aspect does not strongly define coastal landslide susceptibility on the Kaikōura coast. There is some correlation between bedding and aspect within GeolCode 3 along the coast north of the Clarence River mouth (Figure 1) and

in coastal GeolCode 5 where landslide densities are higher. However, hillslopes within the heavily deformed GeolCode 5 may be susceptible to failure regardless of the presence of persistent structural discontinuity. In the heavily jointed rock mass, debris and rock avalanches, the dominant failure mechanism along the Kaikōura coast, can develop along cm- to m-scale





discontinuities (Singeisen et al., 2022) that are not captured by the estimation of larger scale bedding. Furthermore, field investigations along the Kaikōura coast (e.g., Stringer et al., 2021) have shown that many failures from the 2016 earthquake,

particularly in mapped GeolCode 5, occurred as reactivations of pre-existing landslide debris or within Quaternary hillslope deposits that were unlikely to be strongly influenced by bedding orientation. While QMAP (Rattenbury et al., 2006) provides the highest resolution mapping currently available at the required extent for this regional analysis, the mapping resolution is not high enough to sufficiently resolve these materials regionally.

*Fault Zones (Distance to Fault and OFD)* – Bloom et al. (2022a) observed a higher incidence of landslides within the fault zone of ruptured faults from the 2016 Kaikōura earthquake. While there is a slightly higher density of landslides within the OFD zone, there is a lower proportion of OFD area along the Kaikōura coastline as compared to inland hillslopes. Approximately 0.6% of coastal area occurs within the OFD zone of surface fault rupture from the 2016 Kaikōura earthquake while c. 2.5% of inland hillslopes occur within a mapped OFD zone. Landslide source areas that occur within the OFD zone,

account for c. 19% of landslide source area in inland hillslopes but only c. 1% of landslide source area along the coast.

OFD may partially explain the distribution of landslide source areas in inland hillslopes but does little to explain widespread coastal failures or an order of magnitude greater number of coastal landslides. This being said, there is still some ambiguity as to the influence of rock mass deformation from fault zones along the coast that did not rupture significantly in 2016; for

example, the Hope fault which extends just offshore in parallel to much of the north Kaikōura coast. A history of strong ground motion and fault deformation has been shown to progressively decrease rock mass strength and increase landslide susceptibility over multiple earthquakes (Parker et al., 2015; Gischig et al., 2016; Bloom et al., 2022a; Massey et al., 2022). This may result in an increased landslide susceptibility due to amplification of strong ground motion and decreased rock mass strength. While it is possible that damaged rock within the fault zone of the Hope fault results in a higher landslide density in the north Kaikōura

coast, there is also an increase in landslide susceptibility along the coast south of Kaikōura, where faults like the Hundalee are present further offshore (Figure 1). This suggests that the relatively continuous zone of increased coastal landslide density is not solely influenced by fault zones on the Kaikōura coast.

*Anthropogenic modification of slopes (Cutslopes)* – Uplifted shore platforms and marine terraces both north and south of

Kaikōura have been anthropogenically modified by cut and fill slopes to support road and rail infrastructure. Most fill slope failures are too small and are not steep enough to be resolved in this analysis (which considers failures greater than 50 m$^2$ and slopes steeper than 15°). Cutslopes only account for c. 1% of hillslopes along the Kaikōura coastline. Approximately 4% of coastal landslides (63 of 1,621) were found to be in contact with a cutslope near the coast. Even if we consider all failures associated with cutslopes to be a direct result of hillslope modification, this cannot fully explain the higher density of coastal

landslides well beyond anthropogenic influence.



*Precipitation, soil moisture, and enhanced weathering (NDMI)* – NDMI, a proxy for soil moisture, is very slightly higher within coastal hillslopes, particularly in Torlesse greywacke (GeolCode 5), which could indicate increased moisture along the Kaikōura coast one month prior to the earthquake (Figure 7). A high variability in average rainfall observations (NIWA, 2022),

however, makes it difficult to expand this observation out to longer timescales. For instance, it might be expected that increased moisture on the coast would increase chemical weathering rates leading to a reduction in rock mass strength. On a single-event or seasonal timescale, increased NDMI might be a proxy for increased pore water pressure along the Kaikōura coast; however, models suggest that higher NDMI, itself, does not fully explain the distribution of earthquake induced landslides. In Quaternary and Neogene units, and to a lesser extent Paleogene units, lower NDMI is actually a better predictor of landslide

occurrence (Figure 6). NDMI is strongly correlated with vegetation greenness (Table A2) and less vegetation may help to explain some shallow failures. In Lower Cretaceous Torlesse greywacke, NDMI is a comparatively weak predictor of earthquake induced landslides in both inland and coastal hillslopes.

## 5.3 Steep Slopes along the Kaikōura Coast

Based on the distribution of landslide susceptibility features and statistical analysis, the slope feature provides much of the

explanation for increased landslide density along the Kaikōura coastline (Figure 8). Results indicate that average slope (greater than 15°) is slightly (c. 1°) steeper on the coast as compared to inland (Figure B3). While this difference seems small, normalised slope within each GeolCode (Figure 8) reveals substantially steeper slopes within c. 250 to 500 m of the coast particularly within GeolCodes 2, 3, and 5. In many regions, oversteepening results from a combination of uplift and wave action that actively undercuts coastal cliffs (Emery and Kuhn, 1982). In the Kaikōura region, however, most steep coastal

slopes, with the exception of those at Conway Flat (Bloom et al., 2022b), are currently isolated from direct wave action by recent uplift which forms shore platforms. Ages of these uplifted platforms (Howell and Clark, 2022) suggest that this isolation has lasted for several hundred years at the least. Only 16 landslides outside of Conway Flat (c. 1% of landslides) have a direct connection with the ocean. As a result, while rapid uplift of the Kaikōura coast (Ota et al., 1996) contributes to steeper slopes, the contributions of wave erosion to long-term coastal evolution remain somewhat less clear.


During the Kaikōura earthquake, most coastal landslides in GeolCode 5 (c. 83%) occurred partially to wholly within areas affected by past landslides (Table 3), often as reactivations (retrogression) of their head scarps (Figure 2). Similar trends are observed for the distribution of landslides within younger sedimentary units (Table 3). These findings are in line with field observations along the Kaikōura coast following the 2016 earthquake (e.g., Mason et al., 2017; Stringer et al., 2021).


Relict landslides are a common observation in the hillslopes above uplifted shore platforms along the Kaikōura coast, particularly within Torlesse greywacke (GeolCode 5; Figure 2). These relict landslides have left steep, potentially destabilised, headscarps and, in some cases, debris within the body of failures (Stringer et al., 2021). The provenance and timing of these



relict failures is largely unclear but a general lack of deposited material at the base of the hillslopes (Figure 2) suggests that
they may have developed while in contact with an active erosional source, such as rivers or the ocean (Figure 9; Crozier, 2010).

It is possible that uplift is currently out competing active erosion along the Kaikōura coastline resulting in abandoned steep
hillslopes (Figure 9). Hillslope oversteepening can linger following the removal of an active erosional source and increased
landslide susceptibility may remain until hillslopes reach a state of equilibrium with the surrounding landscape (Figure 9;
Crozier, 2010). With coastal uplift rates of c. 2 to 0.5 mm/year (Ota et al., 1996), this could represent up to thousands of years
of increased landslide susceptibility. Without an active erosional source, earthquakes and large rainfall events may
disproportionately contribute to the evolution of these coastal hillslopes (Figure 9). Further investigation would be necessary
to determine the relative contribution of these processes.

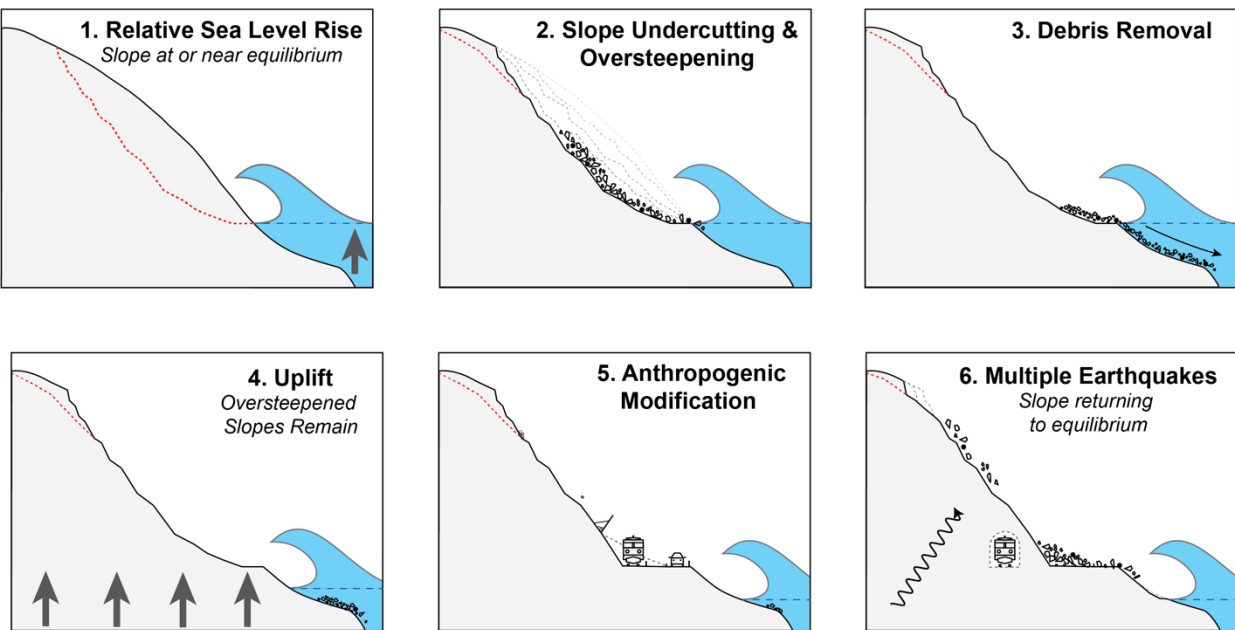

**Figure 9: Conceptual model for the evolution of slope stability along the Kaikōura coast. Hillslopes are oversteepened by active
erosion and debris is cleared from shore platforms by wave action. Following uplift, hillslopes are anthropogenically modified and
earthquakes result in failures within the scars of relict landslides. Terrestrial erosion from earthquakes and rainfall work to bring
the oversteepened hillslope back into equilibrium with the surrounding landscape (Crozier, 2010).**

**5.4 Implications for earthquake induced landslide susceptibility in coastal settings**

Most earthquake induced landslide susceptibility models already rely heavily on geology, strong ground motion, and slope as
predictive features. As such, the findings here support the efficacy of using regionally trained models to characterise earthquake
induced landslide susceptibility on the Kaikōura coast. In other regions, a 'near coast' categorical feature (Figure B1 and B2)



may be a reasonable proxy for other underlying coastal influences. The extent to which such a feature improves larger scale regional models, however, is subject to additional study.


Findings here may be applied to rocky coastlines elsewhere but consideration should be made for other potentially important site-specific conditions that may or may not be incorporated in this investigation. Of particular note, Parker et al. (2015) identified the accumulation of rock mass deformation over multiple earthquakes as a source of landscape preconditioning that results in higher susceptibility to future landslides. Similarly, the scars of relict landslides that occur within the steep hillslopes

of the Kaikōura coastline suggest past susceptibility to failure and a potential accumulation of deformation that is largely unresolved by this analysis (and likely by most regional studies).

Previous investigations (e.g., Marc et al., 2015, 2019; Massey et al., 2022) suggest that increased landslide susceptibility decays to background levels within several years of an earthquake. It may be possible, however, that the factors discussed in this

study, including oversteepened hillslopes, fault deformation, coastal weathering, and repeated earthquake shaking, contribute to an accumulation of stress within the hillslope and, in turn, longer term susceptibility to extreme event failure (Parker et al., 2015). Currently, the detailed rock mass characterisation required to fully investigate the influence of rock mass strength remains largely confined to the site-specific scale. Our understanding of landslide susceptibility along the Kaikōura coast, however, would likely benefit from future studies that attempt to decouple the influence of steep slopes from rock mass

deformation on a regional scale.

As a final note, since the 2016 Kaikōura earthquake, the coastal road and rail corridors north and south of Kaikōura have been fully re-established. In some cases, realignments have been made to address ongoing rockfall and other slope stability concerns (NZTA, 2021). In most cases, however, the road and rail lines have been cleared, repaired, and reopened in their original

alignments (as in panel 5 of Figure 9). Estimates of long-term network resilience were developed shortly after the Kaikōura earthquake and, in part, rely on quantified landslide hazard assessment (Justice et al., 2021). This hazard assessment adopts the established assumption that strong ground motion intensity plays a large role in governing the volume of coseismic landslide debris along the Kaikōura coast (Massey et al., 2019). While our study does not directly address quantified hazard, the results suggest that the distribution of slope angle is generally steeper on the Kaikōura coast compared to inland hillslopes.

These steeper slopes resulted in a high density of coastal landslides during the 2016 Kaikōura earthquake. In future earthquakes, increased coastal landslide susceptibility – the result of steeper slopes along the coast – will expose coastal hillslopes to more landslides than inland hillslopes given the same level of ground motion intensity. The ongoing likelihood of aftershocks and strong ground motion in the Kaikōura region will test the efficacy of mitigation measures installed to reduce risk to people and infrastructure along the coast.





## 6 Conclusions

Distance to the Kaikōura coastline has a substantial influence on the distribution of landslides from the 2016 Kaikōura earthquake. An order of magnitude greater landslide density was observed within 500 m of the coastline (as high as c. 6 %) as compared to 1000 to 3000 m (c. 0.5%). Comparative logistic regression modelling suggests that the same factors, primarily geology, strong ground motion, and slope, define the distribution of landslides in both coastal (within 1 km of the coast) and inland hillslopes (1 to 3 km from the coast). Regional earthquake induced landslide susceptibility models that rely on geology, strong ground motion, and slope as strong predictive features are therefore, likely to account for this increased coastal landslide susceptibility in the Kaikōura region without separate treatment. Along the Kaikōura coastline, hillslopes are generally steeper than those inland, when comparing slopes angles within similar materials. Results suggest that slope angle provides the most explanatory power, and the simplest explanation, for increased coseismic landslide density at the coast. On the Kaikōura coast, most hillslopes are currently buffered from wave action by rapidly uplifted shore platforms; coastal hillslopes host a high density of relict landslides that may have resulted from relatively recent (<1,000 years) coastal erosion. Relict landslides and proportionally steeper hillslopes may maintain a higher coastal landslide susceptibility as a legacy effect within hillslopes out of equilibrium with the surrounding landscape, which may persist for up to 1,000s of years.

## Appendix A – Additional Methods and Data

**Minimum Landslide Size**

Prior to the 2016 Kaikōura earthquake, lidar was available for areas in close proximity (< 1 km) to the Kaikōura coastline. This pre-earthquake lidar coverage likely allowed for more detailed comparison with post-earthquake data. In order to limit any potential bias resulting from differences in the quality of landslide mapping on the Kaikōura coastline in this comparative analysis, we evaluate the size area distribution of earthquake induced landslides mapped by Massey et al. (2020a) in proximity to the Kaikōura coast (Figure A1) using the methods of Malamud et al. (2004). We observed a slightly higher distribution of small failures along the Kaikōura coast with the distribution of failures diverging around 50 m$^2$. While this may represent a real difference in landslide size along the coast, We chose to exclude failures smaller than 50 m$^2$ from the analysis. Because of the small size of failures, this exclusion is unlikely to strongly influence the final results.



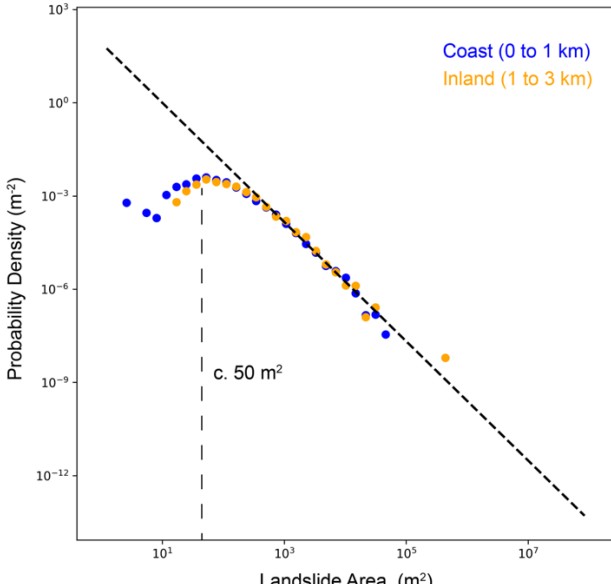

**Figure A1: Size area distribution of landslides (Malamud et al., 2004) from the Kaikōura earthquake induced landslide inventory (Massey et al., 2020a) within 0 to 1 km (Coast) and 1 to 3 km (Inland) of the coastline. The distributions diverge from one another around 50 m2 and we use this as the minimum landslide size threshold.**

**Gridded Landslide Data**

Landslide polygons from the inventory of Massey et al. (2020a) were gridded to the 8 m resolution of the digital elevation model (LINZ, 2022) used to derive topographic landslide susceptibility features. Table 2 shows the percentage of landslide source area in each GeolCode. Table A1 shows the raw number of landslide (1) and non-landslide (0) grid cells used in the analysis.

**Table A1: Landslide (LS) density by GeolCode in all, coast, and inland hillslopes with number of 1 and 0 landslide grid cells.**

| | | GeolCode 1 | GeolCode 2 | GeolCode 3 | GeolCode 5 | Total |
|---|---|---|---|---|---|---|
| **All** | **0** | 165176 | 841438 | 740592 | 1668137 | 3415343 |
| | **1** | 2091 | 6527 | 9942 | 13898 | 32458 |
| | **Total** | 167267 | 847965 | 750534 | 1682035 | 3447801 |
| | **LS Density** | 1.25% | 0.77% | 1.32% | 0.83% | 0.94% |
| **Coast** | | GeolCode 1 | GeolCode 2 | GeolCode 3 | GeolCode 5 | Total |
| | **0** | 71377 | 324791 | 162780 | 419017 | 977965 |
| | **1** | 1838 | 4106 | 5180 | 9315 | 20439 |
| | **Total** | 73215 | 328897 | 167960 | 428332 | 998404 |




| | | GeolCode 1 | GeolCode 2 | GeolCode 3 | GeolCode 5 | Total |
|---|---|---|---|---|---|---|
| | **LS Density** | 2.51% | 1.25% | 3.08% | 2.17% | 2.05% |
| **Inland** | **0** | 93799 | 516647 | 577812 | 1249120 | 2437378 |
| | **1** | 253 | 2421 | 4762 | 4583 | 12019 |
| | **Total** | 94052 | 519068 | 582574 | 1253703 | 2449397 |
| | **LS Density** | 0.27% | 0.47% | 0.82% | 0.37% | 0.49% |


**Landslide Susceptibility Features**

We initially evaluated 25 common predictive features for this analysis (Figure A2). Of the 25 features we narrowed the choice of features using only those features with a variable inflation factor (VIF) score of 10 or less (Table A2). In comparative model training, the best model performance was achieved when using the USGS ShakeMap PGA (Worden et al., 2020) for models 590 of GeolCode 5 and the USGS ShakeMap PGV for all other GeolCodes.

**Table A2: Variable Inflation Factor for features used in this analysis. PGA from the USGS ShakeMap (Worden et al., 2020) is used for GeolCode 5 and PGV from the same model for all other GeolCodes.**

| Feature | VIF— PGV | VIF— PGA |
|---|---|---|
| Curvature | 1.02 | 1.02 |
| Cutslopes | 1.00 | 1.00 |
| Eastness | 1.08 | 1.08 |
| Elevation | 3.22 | 3.22 |
| Fault Distance | 1.84 | 1.86 |
| NDMI | 3.63 | 3.65 |
| Northness | 1.19 | 1.19 |
| OFD | 1.05 | 1.05 |
| Slope | 9.86 | 9.38 |
| Structural Aspect | 3.51 | 3.48 |
| USGS ShakeMap | 6.67 | 6.05 |






**Figure A2: Pearson R2 correlation for common predictive features considered in this analysis. Red features were not included in the final modeling analysis.**





**Deriving the Structural Aspect Feature**

600   To derive a correlation between the dip direction of bedding and topographic aspect, we interpolated structural bedding measurements from the New Zealand QMAP (Rattenbury et al., 2006). We corrected for the 360° direction of dip using the Sin and Cos of dip direction and used the inverse distance weighted (IDW) interpolation method in ArcGIS to produce a continuous estimate of dip direction in each geology type (GeolCode). By subtracting interpolated dip direction from topographic aspect, we get values from -360 to 360° where a value close to -360°, 0°, or 360° represents a close correlation

605 between dip direction of bedding and topographic aspect. To make a continuous range for this analysis, we take the absolute value of the difference (0 to 360), subtract 180 (-180 to 180), and take the absolute value again to arrive at 0 to 180 where 180 represents a high correlation between dip direction and topographic aspect and 0 represents a low correlation.

**Appendix B – Additional Results**

**Full Model**

610 In addition to comparative models of inland and coastal hillslopes, two models were trained using 80% of both inland and coastal data. The first model matches the inland and coastal models included in the main text. The second model includes an additional binary coast feature where inland hillslopes are assigned a value of 0 and coastal hillslopes a value of 1. Both models performed well and were predictive of both inland and coastal landslides in the remaining 20% of data used for testing (Figure B1). Model coefficients from the full model were generally similar to inland and coastal models (Figure B2).

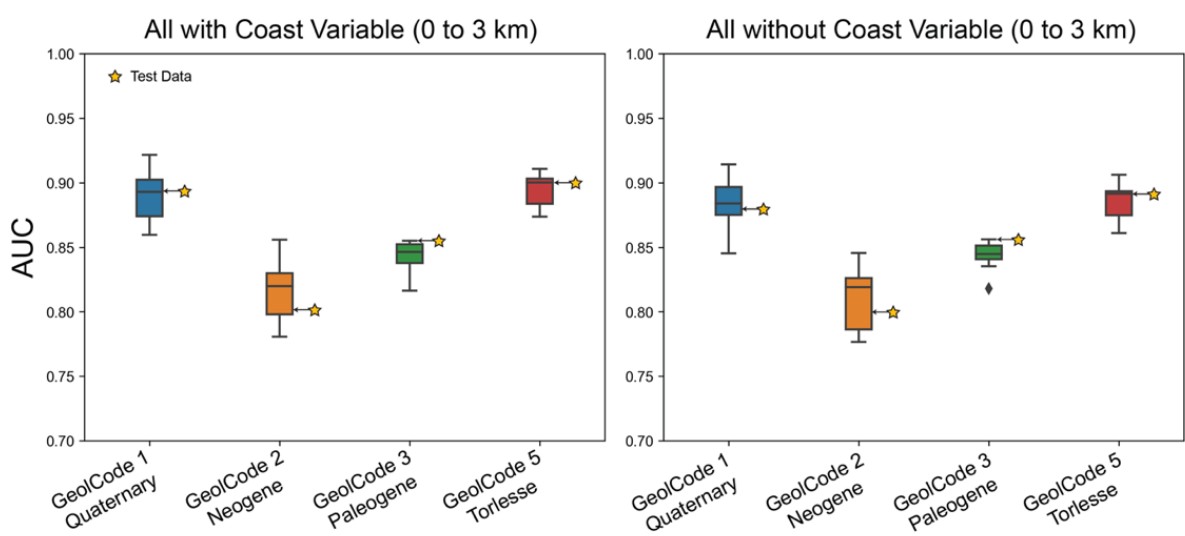


**Figure B1: Logistic regression model performance from models trained on each geology type (GeolCode) in both coastal and inland hillslopes. The model on the left includes a binary coast feature where grid cells within 1 km of the coast are assigned a value of 1 and grid cells within 1 to 3 km of the coast are assigned a value of 0. Model performance is measured by Area Under the Receiver Operator Characteristic Curve (AUC). Each boxplot shows the results of 10-fold cross validation using 80% of the available target**

620 **dataset. Yellow stars represent model performance when applied to the 20% of data withheld from training.**





**Figure B2: Model coefficients for models trained on each GeolCode. Points with associated error bars represent the mean and standard deviation (SD) of the coefficient across 10-cross validations. Colour bars indicate which feature the point is associated with and the model data used to train the model. In cases where no error bar is present, the standard deviation is less than 0.1. Negative coefficients result in a higher weight for small values while positive coefficients result in a higher weight for high values. For example, a negative coefficient for fault distance suggests that there is a higher landslide susceptibility closer to faults while a positive coefficient for slope suggests that a greater slope angle has higher landslide susceptibility. All features are standardised prior to model training allowing for the direct comparison of coefficients within the same model.**



**Coast vs Inland Features and Landslides**

In addition to the analysis presented in the main text we also examined the distribution of predictive features within inland and coastal slopes greater than 15° (Figure B3).

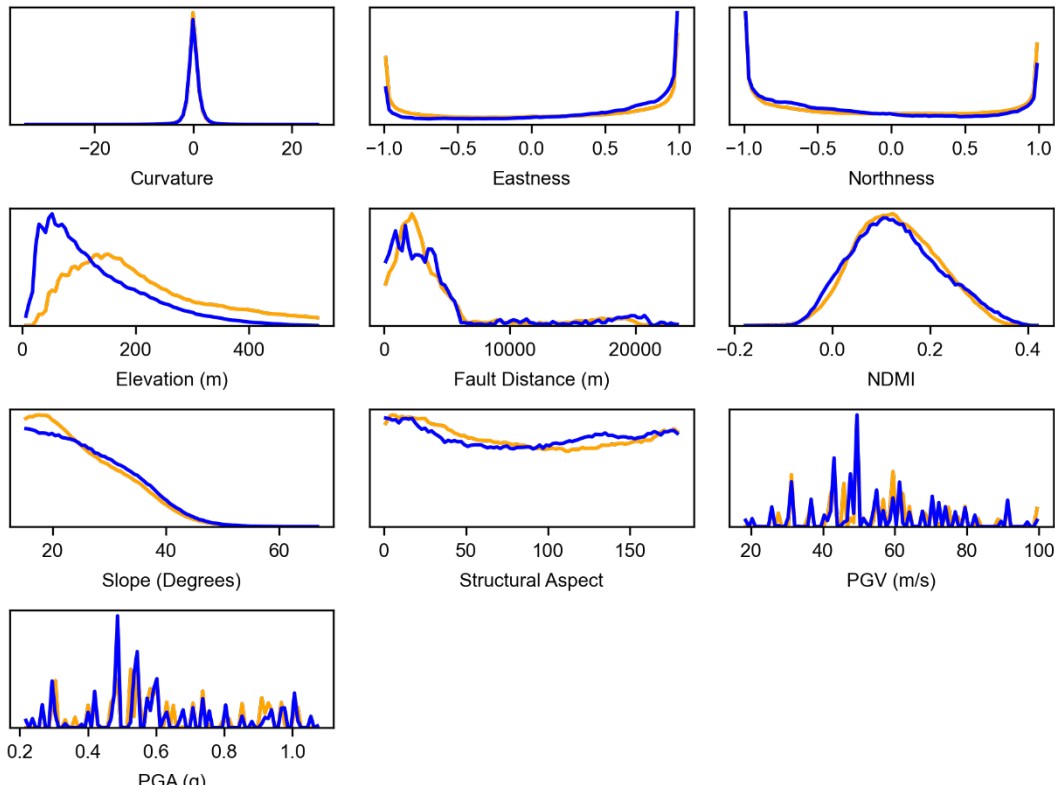

**Figure B3: Distribution of predictive features in all Kaikōura inland and coastal slopes greater than 15°. Inland slopes are represented by orange lines and coastal slopes are represented by blue lines.**


We supplement this analysis by examining the distribution of predictive features within landslide source areas in inland and coastal slopes greater than 15° (Figure B4).





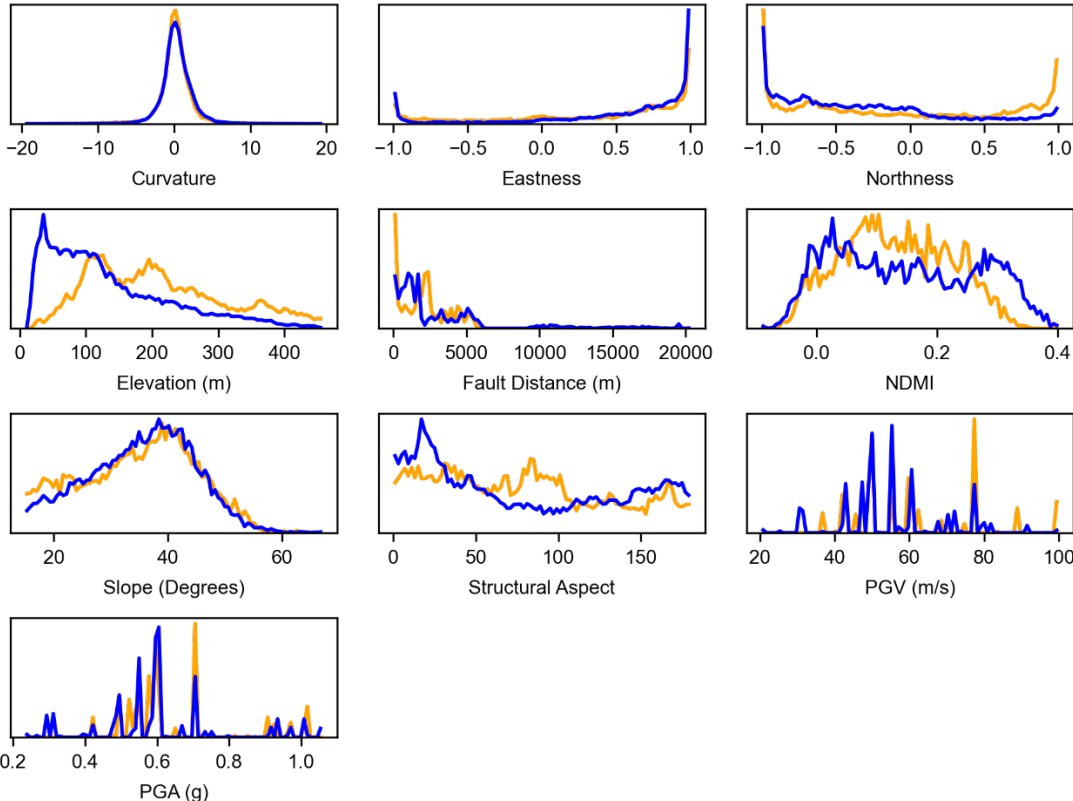

**Figure B4: Distribution of predictive features within Kaikōura earthquake induced landslide source areas on inland and coastal slopes greater than 15°. Inland slopes are represented by orange lines and coastal slopes are represented by blue lines.**

On average, coastal landslides (within 1 km of the coast) are c. 10% larger than inland landslides (between 1 and 3 km from the coast) on slopes > 15° with PGA > 0.2 g (when excluding the Seafront landslide and landslides smaller than 50 m²; Table B1). Given the same filters, there are c. 30% more landslides in coastal hillslopes (Table B1) which make up c. 1/3 of the total study area.

**Table B1: Mean landslide area in inland and coastal hillslopes where slopes > 15°, PGA > 0.2g. Inland estimates exclude the Seafront landslide.**

| Slopes > 15° | Landslide Count | Mean Area (m²) | Standard Deviation (m²) |
|---|---|---|---|
| Inland (1 to 3 km) | 1099 | 783 | 2279 |
| Coast (0 to 1 km) | 1621 | 866 | 2517 |





**Additional Observations**

*Curvature* – Curvature was well distributed and few differences between inland and coastal hillslopes were observed (Figure B3 and B4).

*Aspect (Northness and Eastness)* – Ridgelines and valleys generally trend north-east to south-west in the Kaikōura region (Figure B3) and landslide source areas on both inland and coastal hillslopes occur disproportionately on south to south-east facing hillslopes (Figure B4). Within 1 km of the coastline, a larger proportion of south-east facing hillslopes were observed that generally correlated well with landslide density trends across geology types.

*Elevation* – A steady rise in elevation is observed with distance from the Kaikōura coastline. This rise in elevation, however, does not appear to directly correlate with the landslide density trend that is observed with distance from the coastline.

**Landslide Susceptibility Map**

The models in this study were trained on 80% of available data. In order to make a landslide susceptibility map for this area 665    we have to partially apply the model to data on which it was trained. It should be noted that this significantly limits the applicability of such a map. Figure B5 presents an example of an inland trained landslide susceptibility model in GeolCode 5 applied to both coastal and inland hillslopes. While this model is applied to trained data in inland slopes, coastal data is unseen. In general, the inland trained model captures the extent of coastal landslide susceptibility.



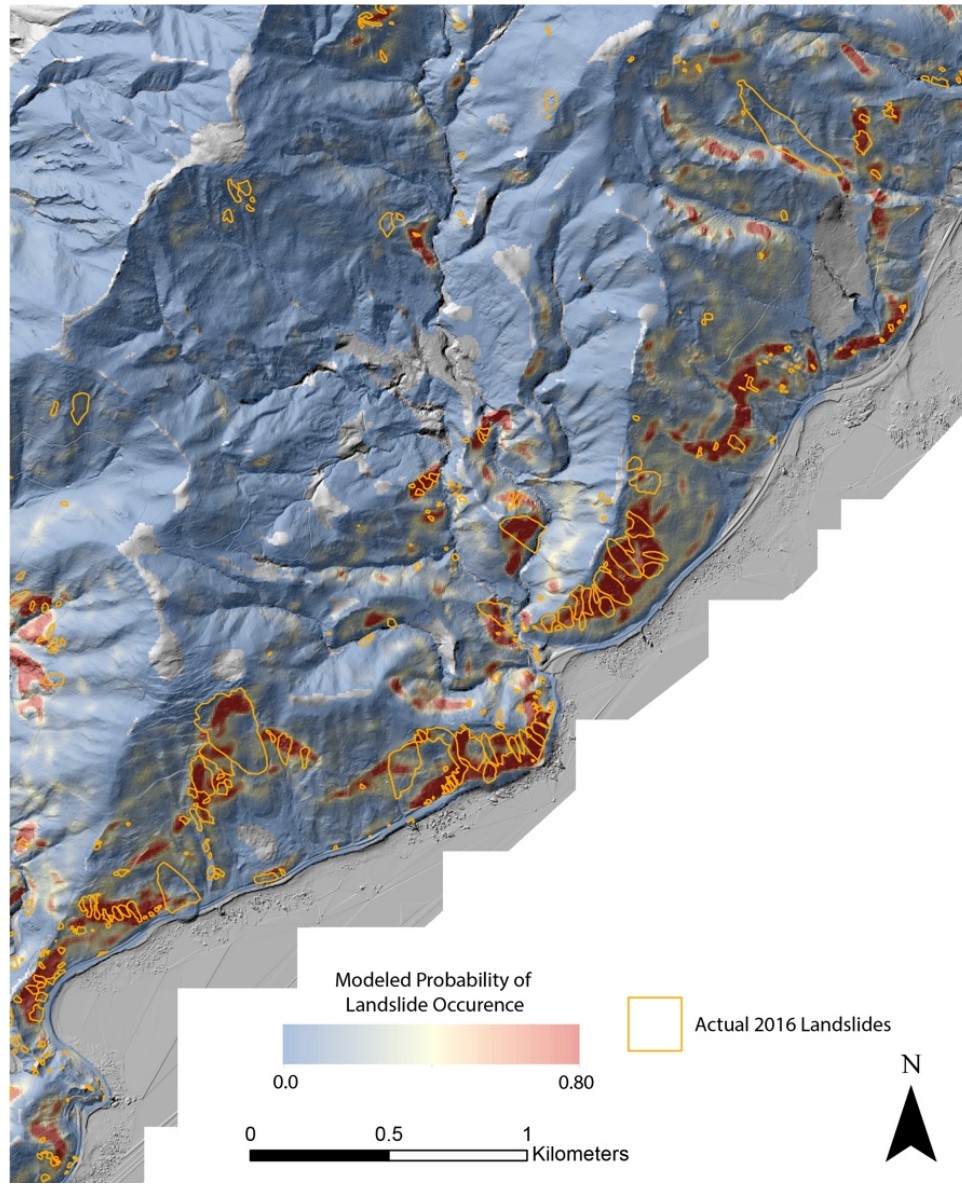

**Figure B5: Results of an inland (1 to 3 km from the coast) trained landslide susceptibility model applied to both inland and coastal hillslopes. The base image is a hillshade of post-earthquake lidar (Massey et al., 2020b) with actual earthquake induced landslides mapped by Massey et al. (2020a). The location is coincident with Figure 2 (location identified in Figure 1).**

**Acknowledgements**

This work was primarily funded by the New Zealand government as part of the Ministry of Business, Innovation and
Employment Endeavour-funded "Earthquake Induced Landslide Dynamics" project with further support from Toka Tū Ake, the New Zealand Earthquake Commission and QuakeCoRE: The New Zealand Centre for Earthquake Resilience. The authors



would like to thank Simon Cox and Dougal Townsend at GNS Science for providing methods and insight on mapping structural domains in the Kaikōura region.

**Author Contributions**

CB: Conceptualization, Methodology, Formal analysis, Investigation, Visualization, Writing – original draft. CS: Methodology, Investigation, Visualization, Writing – review & editing. TS: Supervision, Funding acquisition, Conceptualization, Methodology, Writing – review & editing. AH: Supervision, Methodology, Writing – review & editing. CM: Supervision, Funding acquisition, Writing – review & editing. **DM**: Methodology, Investigation, Writing – review & editing.

**Competing Interests**


The authors declare that they have no conflict of interest.

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
