# Peer review of "Coastal earthquake induced landslide susceptibility during the 2016 Mw 7.8 Kaikōura earthquake, New Zealand"

_EGUsphere, 2022_

## Referee Comment (RC2)

[referee-annotated manuscript omitted]

---

## Author Comment (AC1)

[Figure]

*Figure A3. Relationship between LINZ 8 m DEM (LINZ, 2021a) and a 2012 Lidar DEM (LINZ, 2021c) in the top plot. The bottom plot shows the relationship between topographic slope derived from the same two datasets.*

---

## Author Response (AR1)

*Thank you for your review of this manuscript. We have revised the manuscript based on reviewer comments and believe these revisions have significantly improved the clarity of the text. We look forward to your further consideration. Our responses to individual reviewer comments are included below.*

*Dear Reviewer 1,*

*Thank you for your comments. We have responded to your suggestions below (our responses are in italics). We have made some revisions to the manuscript based on your comments that we believe will help to clarify our work.*

GENERAL COMMENTS

The paper investigates the differences between coastal and inland landslides and evaluates probabilistic landslide models regarding their prediction performance for regional assessments. The research scope and methodology are clear, and the results are well presented. However, the implications of the findings remain a bit vague. The discussion suggests that there are no significant differences between the coastal and inland trained model, raising the question on how the results of the paper can contribute to future landslide assessments. The paper could benefit from highlighting the value of the findings and elaborating on the scope for future research.

*R. The final discussion section in the manuscript, currently Section 5.4, highlights the implications of this study and areas for future research. We discuss three points in detail:*

*1. Lines 515 to 519: That "the findings here support the efficacy of using regionally trained models to characterise earthquake induced landslide susceptibility on the Kaikōura coast absent additional predictive features." This is the primary finding of the manuscript and supports the use of regional landslide susceptibility assessments. In general, in the Kaikōura region, regional models do not need to separately consider coastal regions despite high landslide incidence.*

*2. Lines 521 to 526: That the findings here may be applied to coastlines elsewhere but should consider other potential site-specific controls including accumulated rock mass deformation. At present, it is not possible to resolve rock mass deformation on a regional scale and this limits the findings of this and all regional landslide susceptibility studies.*

*3. Lines 537 to 549: That re-development of the Kaikōura coastal road and rail corridor, as documented by Justice et al. (2021), was undertaken under an assumption that stronger ground motion near the coast resulted in a higher density of failures. This study, however, suggests that steep slopes may have played a more dominant role than ground motion. Under the same ground motion conditions, the Kaikōura coast is likely to have more failures than inland slopes. Future studies may consider revising and re-evaluating future infrastructure risk on the Kaikōura coast.*

*To ensure that that the implications of this work are better supported and made clear to future readers, we have made several changes to the introduction and discussion sections as follows:*

*First, in the introduction on line 52 we have added text to explicitly highlight the purpose of the analysis: "In this study, we test whether a landslide susceptibility model trained on landslides from inland hillslopes captures the distribution of coastal landslides from the 2016 Kaikōura earthquake*

*in New Zealand. The purpose of this test is to determine if existing variables can explain the difference between inland and coastal landslide densities, or if future landslide susceptibility models should consider additional coastal specific variables."*

*Second, we have renamed and revised Section 5.1 to be "Section 4.5 Summary of Results." As the reviewer suggests, this section highlights that there are few significant differences between coastal and inland trained models. By moving Section 5.1 into the results, Section 5.2, which specifically addresses reasons for a higher landslide density along the Kaikōura coast, leads the discussion.*

*Third, we have renamed and revised Section 5.3 to be "Section 5.2 Conceptual Model Relating Coastal Hillslope Morphology and Landslide Susceptibility to Geomorphic History." This modification has been supported by additional text and revisions in this section which clarify our conceptual model for Geomorphic Evolution of the Kaikōura coast. These changes provide a clearer transition into the final discussion section (current Section 5.4) addressing implications and areas for future work. As an example of these modifications, lines 502 to 508 now read: "In our conceptual model, wave action leads to cliff collapse and landsliding while the ocean is in direct contact with the hillslope (Fig. 10-1 and 10-2). Once uplift or relative sea level fall occurs, hillslopes are buffered from wave action at the toe but remain over-steepened, particularly within upper slopes where relict landslide headscarps are present (Figure 10-3 and 10-4). In the case of the Kaikōura region, uplifted wave cut platforms provide ideal locations for transportation infrastructure (Fig. 10-5). While modification of lower hillslopes may not result in substantially increased landslide susceptibility on a regional scale, over-steepened upper hillslopes may remain highly susceptible to coseismic failure over multiple earthquake events (Figure 10-6; Rault et al., 2018; Singeisen et al., 2022). Increased susceptibility is likely to persist until the hillslope reaches a state of relative equilibrium with the surrounding landscape or until active erosion recurs at the base of the slope (Crozier, 2010). With coastal uplift rates of c. 2 to 0.5 mm/year (Ota et al., 1996) along the Kaikōura coast, oversteepened slopes that are not currently being eroded by wave action could still represent up to thousands of years of increased landslide susceptibility (Singeisen et al., in Review). Without an active erosional source, earthquakes and large rainfall events may disproportionately contribute to the geomorphic evolution of these coastal hillslopes (Figure 10)."*

*Finally, we have slightly modified the implications section, (current) Section 5.4. Lines 515 to 519 now read: "Most earthquake induced landslide susceptibility models already rely heavily on lithology, strong ground motion, and slope as predictive features. As such, the findings here support the efficacy of using regionally trained models to characterise earthquake induced landslide susceptibility on the Kaikōura coast absent additional predictive features. In the Kaikōura region, a 'near coast' categorical feature (Figure B1 and B2) does not substantially improve model prediction. In other regions, such a feature may serve as a reasonable proxy for other underlying coastal influences, but this is subject to additional study." To highlight additional needs for future work we have also modified lines 533 to 535 to read: "Our understanding of coastal landslide susceptibility would benefit from future studies that attempt to decouple the influence of steep slopes from rock mass deformation on a regional scale."*

SPECIFIC COMMENTS

Line 46 on page 2: What kind of landslide models are we talking about here? If we are looking at probabilistic models which are trained on landslide inventories, and if "a significantly higher landslide density was observed on coastal hillslopes as compared to inland hillslopes", should we not

expect that those models reflect the landslide distribution across these areas? Perhaps, you could provide a bit more context on the landslide models regarding how they are biased towards inland hillslopes.

*R. To clarify, we have modifed line 49 to 52 to read: "In several cases (e.g., Griggs and Plant, 1998; Collins et al., 2012; Massey et al., 2018) a significantly higher landslide density was observed on coastal hillslopes as compared to inland hillslopes, but this has yet to be considered in most probabilistic landslide susceptibility assessments. Given the potential influence of increased precipitation, weathering, and soil moisture along coastlines, it is possible that regional earthquake induced landslide susceptibility models, primarily trained on more abundant inland hillslopes, may not effectively predict coastal landslide distributions."*

Line 56 on page 2: "No clear physical control on landslide density was identified although several hypotheses were explored" – What hypotheses were explored? Further explanation would be helpful here.

*R. We have revised line 56 to read "...several hypotheses were presented including differing slope geometry resulting in ground motion amplification along the Kaikōura coast and reduced rock mass strength."*

Line 209 on pages 8 (Table 1): The LINZ 8 m DEM is used to calculate different features. LINZ data service states that the "main criterion in its production was the detailed and accurate depiction of natural landforms. It is therefore suitable primarily for cartographic visualisation. Because it was created by the interpolation of 20m contours with post-processing and filtering it is not suitable for terrain analysis." (https://data.linz.govt.nz/layer/51768-nz-8m-digital-elevation-model-2012/metadata/). What are your thoughts on this? Have you thought about potential issues in the outcome using the 8 m DEM data? I used the 8 m DEM and the 25m DEM by LRIS, which was derived from the same 20m contour lines, to calculate slope and other hydrology related features, and I encountered several issues as the results differ significantly in some places. So it would be interesting to know what your reasons are for using the 8 m DEM, especially given the fact that other variables such as the NDMI have a lower resolution.

*R. While the LINZ 8 m DEM may not represent the most ideal dataset with which to conduct terrain analysis, it is the highest resolution elevation dataset available with consistent coverage of the Kaikōura region prior to the 2016 Kaikōura earthquake. A number of DEMs have been derived from the New Zealand 20 m contours but, to the best of our knowledge, there is no dataset in New Zealand with consistent coverage that is widely accepted or considered substantially more trustworthy than the LINZ 8 m DEM. Given the relatively small size of many coastal landslides, using a coarser DEM, for example the Shuttle Radar Topography Mission (SRTM) digital elevation model at 30 m, would likely result in underprediction of coastal landslides and limit our ability to make robust claims about differences in landslide susceptibility between inland and coastal slopes. This is the same reason we upsample coarser resolution data, for example NDMI from 30 m Landsat data, to match the 8 m grid cell size.*

*The regional nature of this assessment should reduce concern about local irregularities in the underlying DEM. To examine this assumption, we downsampled and compared limited pre-earthquake lidar collected along the Kaikōura coast in 2012 (LINZ, 2021c) with the coincident LINZ 8 m DEM (LINZ, 2021a). Limited lidar coverage extends to c. 500 m inland along much of the*

*Kaikōura coastline and accounts for c. 30% of the study area included in this analysis. We find a strong one-to-one correlation between the two datasets (Figure A3 is included in the appendix of the revised manuscript). On the regional scale of this analysis, the LINZ 8 m DEM and its derivations are, indeed, largely characteristic of terrain conditions identified within lidar. We have added text to the end of line 212 which reads: "While the LINZ 8 m DEM has known limitations for local terrain analysis, on a regional scale, it captures the majority of terrain characteristics (additional information in Appendix A) and is one of the few datasets with appropriate coverage and resolution for our analysis."*

Line 242 on page 10: It is stated that the ROC AUC was used to "demonstrate the relatively high performance of all trained models". Considering the high class imbalance (around 99% of the sample are negative cases), is it not possible that the AUC is biased towards the majority class?

*R. You are likely correct that AUC is biased towards the majority class. In the case of this study, we are not attempting to optimize model performance and we simply use ROC/AUC to establish that these models are 1) relatively robust and 2) that inland models fairly reliably predict coastal landslide distributions. Visual inspection of model results (e.g. former Figure B5, which has been revised to Figure 6) confirm that these models are indeed fairly robust and that our relatively strong model performance using AUC is not just a result of imbalanced data.*

*We have added Figure B5 into the main text and include the following text on line 317: "Visual inspection of modelled landslide probability when applying the inland trained model to coastal hillslopes also suggests relatively strong model performance (Figure 6). We generally observe higher landslide probability where coastal landslides (not included in model training) occurred during the Kaikōura earthquake."*

Line 370 on page 18: What could be the reasons for the similar performance and coefficients? Could this be related to the observational data (Kaikōura)? Or could the conditions between the coast and the inland terrain be too similar, so developing a coast specific model does not add any (measurable) value?

*R. It is our interpretation that similar model coefficients and good model performance for inland trained models applied to coastal hillslopes result from very little difference in landslide susceptibility between the two regions. As demonstrated in Figure 8 (revised to Figure 9) and discussed in current Section 5.2, observed landslide density and modelled landslide susceptibility increase coincident with an increase in topography closer to the Kaikōura coast providing an explanation for increased landslide density absent a difference in the factors controlling landslide susceptibility. As we note on lines 521 to 522, different study areas could have variable conditions that influence landslide density along the coastline.*

Line 481 on page 22 (related to the previous comment): Considering that the slope is the driving variable in the prediction of landslides and considering that the slope along the coast is only 1° steeper on average, it does not seem surprising that a model trained on costal landslides leads to similar results. The discussion and conclusion are a bit vague on how the findings can contribute to future assessments. Adding a coastal factor to the model is an interesting idea. However, the ROC AUC charts (Figure B1) suggest that the prediction performance does not benefit from that factor. It would be helpful to provide details on what future research needs to do in order to better understand the differences between the coastal and inland landslide susceptibility.

*R. As described previously, current Section 5.4 outlines both the primary findings of this study and the future research needed to better understand differences between coastal and inland landslide susceptibility. Existing regional landslide susceptibility models, which include slope as an important predictive feature, are likely to capture increased landslide density associated with steeper slopes along the Kaikōura coast. The influence of heterogeneous rock mass deformation within lithologic units remains largely uncharacterized and underrepresented in models representing an important avenue for further study.*

Line 670 on page 34 (Figure B5): Have you plotted a probability map using the inland model for this region? It would be interesting to see where the differences are.

*R. Figure B5 (to be revised to Figure 6) represents the results of applying the inland trained model to both inland and coastal slopes.*

TECHNICAL CORRECTIONS

Line 71 on page 3 (Figure 1): The caption contains a lot of information that could be shortened, for example, by removing "The labelled MFS or Marlborough Fault System is north of the Hope fault and the NCD or North Canterbury Domain is south of the Hope fault." This could be mentioned in the text.

*R. We have modified the caption of Figure 1 slightly to reduce text and we include some additional information on the MFS and NCD in revised text on line 65: "The earthquake, which caused complex surface deformation along c. 110 km of coastline (Clark et al., 2017), ruptured faults of both the North Canterbury Domain (NCD) and the Marlborough Fault System (MFS) tectonic domains which transfer plate motion from the Hikurangi Subduction Zone in the north to the transpresive Alpine Fault in the south (Figure 1, Litchfield et al., 2018)."*

Line 107 and 116 on page 5: Different units are used for year ("$y^{-1}$" vs "$yr^{-1}$").

*R. This has been resolved.*

Line 146 and 174 on page 7: Assuming that "Land Information New Zealand Topo50 coastline" and "1:50k Topo50 New Zealand Coastline" refer to the same dataset, choose one description and use consistently throughout the paper.

*R. This has been resolved.*

Line 211 on page 9: Space between the number and the unit is missing in "8m" and "20m". Also applies to some cases in Table 1.

*R. This has been resolved.*

Line 291 (Table 2) and 297 (Table 3): Is there a reason why the numbers are presented in a table and not a chart (e.g., bar chart)?

*R. We believe that reliably providing this information in a chart form would require 14 plots and this would likely become more confusing than the current table format.*

Line 799 on page 38: Please provide further information of the data references, for example, name of the dataset, date of access etc.

*R. This has been resolved.*

Line 901 on page 41: Similar to previous comment. Please provide further information on the ground shaking data used for the assessment.

*R. This information has been included in Table 1.*

*Dear Reviewer 2,*

*Thank you for your comments. We have responded to your suggestions below (our responses are in italics). We have made some revisions to the manuscript based on your comments that we believe will help to clarify our work.*

General Comment:

The manuscript investigates the influence of controlling factors on earthquake-induced landslides in coastal hillslopes during the Kaikoura earthquake. The paper is very well presented, methods are clear and sound. The results however are not too impressive, in the sense that it does not appear to be a clear answer to the question of why there is a much higher landslide density closer to the coast, though the main controlling factors of landslides are identified. I suggest a bit more discussion on the proposed key factors, citing literature from other case studies and proposing avenues for further research following this line. In terms of structure of the paper, I suggest including a brief seismotectonic setting in chapter 2, and adding the susceptibility map to the main body of the paper. These and other suggestions are included in the attached file.

*R. Section 5.2 discusses several potential factors contributing to the landslide distribution. Steeper slopes along the Kaikōura coast provide the clearest explanation for increased landslide density and this is one of the primary findings of this work.*

*To ensure that the controlling influence of slope along the Kaikōura coast is clear to future readers, we have revised lines 400 to 405 to read: "Slope – Model coefficients and jackknife models (\*Current\* Figure 6 and 7) suggest that slope is one of the most important features determining the distribution of landslides from the Kaikōura earthquake in both inland and coastal slopes. Massey et al. (2018) noted a lower overall distribution of slope near the coast in the Kaikōura region, however, when hillslopes below 15 are excluded, we observe a slightly higher average slope (c. 1) within 1 km of the coast as compared to 1 to 3 km inland (Figure B3). While this difference may seem small, the variation in slope with distance from the coast largely mirrors modelled landslide susceptibility and landslide density trends across geology types (\*Current\* Figure 8). Steeper slopes, predominantly*

*those occurring within c. 500 m of the coastline (\*Current\* Figure 8), appear to have a primary control on increased coseismic landslide density in proximity to the Kaikōura coastline."*

*As we discuss in Section 5.3, it is not currently possible to characterise reduced rock mass strength (from a combination of physical and chemical weathering) on a regional scale. This is the largest limitation in our analysis and represents an important avenue for future research. These limitations do not prevent us from testing our primary research question "do inland trained models capture coastal landslide susceptibility in the Kaikōura region?" As slightly revised on line 516: "the findings here support the efficacy of using regionally trained models to characterise earthquake induced landslide susceptibility on the Kaikōura coast absent additional predictive features."*

*Additional comments are discussed in detail below.*

Line-by-line:

Line 50: I would not generalize increased precipitation as a coastal characteristic, in many places it increases inland with higher relief.

*R. We agree. We have modified the text on line 50 to read: "Given the potential influence of increased precipitation, weathering…"*

Line 61: I miss a Tectonic Setting section, to get the a tectonic context that helps to understand better the fault systems and uplift/subsidence that are mentioned in this chapter

*R. As written, Section 2.1 provides a brief introduction to the 2016 Kaikōura earthquake and landslide distribution while Section 2.2 introduces the coastal geologic setting and longer-term coastal uplift within the region. In the revised manuscript we have included some additional basic information on the tectonics of the Kaikōura region on line 65: "The earthquake, which caused complex surface deformation along c. 110 km of coastline (Clark et al., 2017), ruptured faults of both the North Canterbury and the Marlborough Fault System tectonic domains which transfer plate motion from the Hikurangi Subduction Zone in the north to the transpresive Alpine fault in the south (Figure 1, Litchfield et al., 2018)."*

*We appreciate that some readers may be interested in a more detailed overview of the tectonic context of the region, however, given the context of this analysis, we do not believe that an independent section is warranted here. This work focuses on the applicability of inland trained landslide susceptibility models to coastal landslide susceptibility and providing improved explanation for increased landslide density along the Kaikōura coastline. Other publications, for example Litchfield et al., (2018) and Clark et al., (2017) provide much more detail on regional tectonics and our liberal use of these citations in Sections 2.1 and 2.2 should direct interested readers to the appropriate additional resources.*

Line 66: what is the mechanism of these faults (long-term and for the earthquake)?

*R. As discussed in the comment above, we direct readers to Litchfield et al. (2018) and Clark et al. (2017) for additional information on faulting, regional tectonics, and coastal uplift in the Kaikōura region.*

Line 109: specify if upward or downward vertical movements for the earthquake along the coast, or are all subsidence?

*R. Coastal vertical displacement during the 2016 Kaikōura earthquake varied from c. -2.5 to 6.5 m. (Line 109)*

Line 143: does the landslide inventory distinguish between source area and deposit? If not, how do you recognize the source area?

*R. Mapped source area polygons were retrieved from version 2.0 of the 2016 Kaikōura earthquake induced landslide inventory by Massey et al., 2020a (Line 141). These do not include landslide deposits.*

Line 166: indicate what tool(s) you used to identify pre-existent landslides and pre-earthquake topography.

*R. We reviewed the high resolution pre- and post-event digital elevation models and orthophotographs produced by Massey et al. (2020a) to create the Kaikōura earthquake induced landslide inventory. We have clarified on line 160 that we visually inspected this imagery: "To supplement the landslide inventory, we visually reviewed…"*

Line 205: should define VIF to fully understand this important step

*R. We have added the following sentence defining VIF at the end of line 207: "VIF, defined as: $VIF= 1/(1-R_i^2)$ is an assessment of the linear relationship between any individual feature and all other potential features ($R_i^2$) (Kutner et al., 2004). Excluding collinear features ensures more representative model weighting, Removal limits the influence of collinear features, generally improving improves model performance, and maintaining maintains model explainability."*

Line 297 (Table 3): what is this table?

*R. Table 3 represents the distribution of landslides in relation to past failures observed within the same hillslope. We have modified the caption to read: "Earthquake induced landslide activity in relation to past failures"*

Line 418: there some interesting examples of strong motion effects on landslide susceptibility in the chapters of Massey et al and Sepulveda in the book Coseismic Landslides (Towhata et al, eds) by Springer 2022, that may be worth to cite for more discussion on these issues

*R.  Citations to both chapters have been added to the revised manuscript.*

Line 467: very slightly higher? how much is that?

*R. We have revised Figure 9 (former Figure 8) to show the distribution of NDMI for each GeolCode and have revised line 467 to read: "NDMI, a proxy for soil moisture, is generally similar within coastal and inland hillslopes (Figure B3). In Torlesse greywacke (GeolCode 5), NDMI is on average 0.17 for coastal hillslopes and 0.13 for inland hillslopes which could indicate increased moisture*

*along greywacke portions of the Kaikōura coast one month prior to the earthquake (Figure 9 and Figure B3)."*

Line 508: may be some influence of higher weathering due to coastal fog/mist close to the coast? That's sometimes the case in uplifted coastlines with rocky cliffs or steep slopes that stop low clouds. Just an idea, maybe it does not apply, not sure if this is reflected in the NDMI

*R. This is a possibility, unfortunately the rock mass strength data necessary to decouple the long-term influence of increased moisture and chemical weathering from other rock mass deformation is not available on a regional scale. We have modified line 469 slightly to read: "A high spatial variability in average rainfall observations (NIWA, 2022), however, makes it difficult to expand this observation out to longer timescales. It might be expected that increased rainfall and moisture on the coast would increase chemical weathering rates leading to a reduction in rock mass strength, but it is not currently possible to characterise these influences on a regional scale."*

Line 530: I would add topographic amplification

*R. Topographic amplification has been added to this list.*

Line 669: I suggest adding the map as a figure in the main body, it helps to visualize the results in charts from other figures

*R. We have moved this example of modelled probability (former Figure B5) into the main body of the text as the new Figure 6.*